# On the Universality of Langevin Diffusion for Private Euclidean (Convex) Optimization

## Abstract

In this paper, we revisit the problem of differentially private empirical risk minimization (DP-ERM) and differentially private stochastic convex optimization (DP-SCO). We show that a well-studied continuous time algorithm from statistical physics, called *Langevin diffusion* (LD), simultaneously provides optimal privacy/utility trade-offs for both DP-ERM and DP-SCO, under $\varepsilon$-DP, and $(\varepsilon, \delta)$-DP both for convex and strongly convex loss functions. We provide new time and dimension independent uniform stability properties of LD, with which we provide the corresponding optimal excess population risk guarantees for $\varepsilon$-DP. An important attribute of our DP-SCO guarantees for $\varepsilon$-DP is that they match the non-private optimal bounds as $\varepsilon \to \infty$.

## 1 Introduction

Over the last decade, there has been significant progress in providing tight upper and lower bounds for differentially private empirical risk minimization (DP-ERM) (Chaudhuri et al., 2011; Kifer et al., 2012; Bassily et al., 2014; Song et al., 2013; McMahan et al., 2017; Smith et al., 2017; Wu et al., 2017; Iyengar et al., 2019; Song et al., 2020; Chourasia et al., 2021) and differentially private stochastic optimization (DP-SCO) (Bassily et al., 2019; Feldman et al., 2020; Bassily et al., 2020; Kulkarni et al., 2021; Gopi et al., 2022; Asi et al., 2021b), both in the $\varepsilon$-DP setting and in the $(\varepsilon, \delta)$-DP setting[1]. While we know tight bounds for both DP-ERM and DP-SCO in the $(\varepsilon, \delta)$-DP setting (Bassily et al., 2014; 2019), the space is much less understood in the $\varepsilon$-DP setting (i.e., where $\delta = 0$). First, to the best of our knowledge, tight DP-SCO bounds are not known for $\varepsilon$-DP. In this paper when we say a bound is tight for any problem, we implicitly always expect the bound to reach the optimal non-private bound (including polylogarithmic factors) for the same task as $\varepsilon \to \infty$. Second, the algorithms for both DP-ERM and DP-SCO in the $\varepsilon$-DP setting are inherently different from the $(\varepsilon, \delta)$-DP setting. While all the algorithms in the $(\varepsilon, \delta)$-DP setting are based on DP variants of gradient descent (Bassily et al., 2014; 2019; Feldman et al., 2020; Bassily et al., 2020), the best algorithms for $\varepsilon$-DP are based on a combination of exponential mechanism (McSherry & Talwar, 2007) and output perturbation (Chaudhuri et al., 2011). Third, we know that as we move from $\varepsilon$ to $(\varepsilon, \delta)$-DP, for convex problems, we gain a polynomial improvement in the error bounds in terms of the model dimensionality, $p$. It is unknown if such an improvement is even possible when the loss functions are non-convex. In this work, we close these gaps in our understanding of DP-ERM and DP-SCO via the following contributions.

---

[1]We only focus on $\ell_2$-Lipschitz losses and the constraint set is bounded in the $\ell_2$-norm; the non-Eucledian setting (Talwar et al., 2015; Asi et al., 2021a; Bassily et al., 2021) are beyond the scope of this work.

1. We provide a unified framework for DP-ERM/DP-SCO via an information theoretic tool called *Langevin diffusion* (Langevin, 1908; Lemons & Gythiel, 1997) (LD) (defined in eq. (1) and eq. (2)), which under appropriate choice of parameters interpolates between optimal/tight utility bounds for both DP-ERM and DP-SCO, both under $\varepsilon$ and $(\varepsilon, \delta)$-DP.

2. We provide tight DP-SCO bounds for both convex and strongly convex losses under $\varepsilon$-DP. To achieve these bounds, we show uniform stability of the exponential mechanism on strongly convex losses (which was to the best of our knowledge unknown prior to our work). Notably, our proof of uniform stability is almost immediate given the unified framework from item 1.

3. We provide a lower bound showing that if the loss functions are non-convex, it is not possible to obtain any polynomial improvement in the error in terms of the model dimensionality when we shift from $\varepsilon$-DP to $(\varepsilon, \delta)$-DP. This is in sharp contrast to the convex setting.

4. Along the way we provide a set of results, which may be of independent interest:
   (a) A simple Rényi divergence bound between two Langevin diffusions running on loss functions with a bounded gradient difference.
   (b) For strongly convex and smooth losses, a Rényi divergence bound between two Langevin diffusions that approaches the Rényi divergence between their stationary distributions.
   (c) Improved analyses for $\varepsilon$-DP-ERM: For strongly convex losses we improve the bound by log factors via a better algorithm, and for non-convex losses we remove the assumption of Bassily et al. (2014) that the constraint set contains a ball of radius $r$.
   (d) A last-iterate analysis of Langevin diffusion as an optimization algorithm, using continuous analogs of techniques in Shamir & Zhang (2013). To the best of our knowledge, this is the first analysis tailored to a continuous-time DP algorithm.

Our work initiates a systematic study of DP continuous time optimization. We believe this may have ramifications in the design of discrete time DP optimization algorithms analogous to that in the non-private setting. In the non-private setting, continuous time dynamical viewpoints have helped in designing new algorithms, including the celebrated mirror-descent (see the discussion in Chapter 3 of Nemirovskij & Yudin (1983)) and Polyak's momentum method (Polyak, 1964), and understanding the implicit bias of gradient descent (Vardi et al., 2022). Extending the dynamical viewpoint to private optimization would help us understand the price of privacy for the convergence rate of private optimization. Further, it is known that privacy helps in generalization and implicit bias underlies the generalization ability of (non-private) machine learning models trained by stochastic gradient descent. Taking the dynamical viewpoint would help us understand if there is a more fundamental reason for the generalization ability of differential privacy.

In the rest of this section, we elaborate on each of these conceptual contributions, and alongside highlight the technical advances that were required. At the end, we provide comparison to most relevant works. We defer a broader comparison to other works in the area to Appendix A.

**Problem description:** Consider a data set $D = \{d_1, \ldots, d_n\}$ drawn from some domain $\tau$ and an associated loss function $\ell : \mathcal{C} \times \tau \to \mathbb{R}$, where $\mathcal{C} \subset \mathbb{R}^p$ is called the *constraint set*. The objective in DP-ERM is to output a model $\theta^{\mathsf{priv}}$ while ensuring differential privacy that approximately minimizes the *excess empirical risk*, $\mathsf{Risk}_{\mathsf{ERM}}(\theta^{\mathsf{priv}}) = \frac{1}{n} \sum_{i=1}^{n} \ell(\theta^{\mathsf{priv}}; d_i) - \min_{\theta \in \mathcal{C}} \frac{1}{n} \sum_{i=1}^{n} \ell(\theta; d_i)$.

For brevity, we will refer to $\frac{1}{n} \sum_{i=1}^{n} \ell(\theta; d_i)$ as $\mathcal{L}(\theta; D)$, or the empirical loss. For DP-SCO, we assume each data point $d_i$ in the data set $D$ is drawn i.i.d. from some distribution $\mathcal{D}$ over the domain $\tau$, and the objective is to minimize the *excess population risk* given by $\mathsf{Risk}_{\mathsf{SCO}}(\theta^{\mathsf{priv}}) = $

$\mathbb{E}_{d \sim \mathcal{D}} \left[ \ell(\theta^{\mathsf{priv}}; d) \right] - \min_{\theta \in \mathcal{C}} \mathbb{E}_{d \sim \mathcal{D}} \left[ \ell(\theta; d) \right]$. In other words, the goal of DP-ERM is to output $\theta^{\mathsf{priv}}$ that minimizes the average loss on $D$ while the goal of DP-SCO is to output a $\theta^{\mathsf{priv}}$ that performs well on the 'unseen' data sampled from $\mathcal{D}$. It is easy to see that DP-SCO is a stronger requirement.

Throughout the paper, we make two standard assumptions in differentially private optimization: (i) the loss function $\ell(\theta; d)$ is $L$-Lipschitz w.r.t. $\ell_2$-norm, i.e., $\forall \theta \in \mathcal{C}, d \in \tau, \|\nabla_\theta \ell(\theta; d)\|_2 \leq L$ and (ii) constraint set has bounded diameter. W.l.o.g., we assume the loss functions are twice continuously differentiable within the constraint set $\mathcal{C}$ – if not, we can ensure this by convolving the loss function with finite variance Gaussian kernel (Feldman et al., 2018). Depending on the problem context, we make additional assumptions like $m$-strong convexity, i.e., $\nabla_\theta^2 \ell(\theta; D) \succeq m\mathbb{I}$, and $M$-smoothness, i.e., $\nabla_\theta^2 \ell(\theta; D) \preceq M\mathbb{I}$, where $A \preceq B$ denotes that $B - A$ is a positive semidefinite matrix. We drop the subscript $\theta$ when it is clear from the context. We provide notational details in Appendix A.

**Langevin Diffusion (LD):** We will start with the Langevin Diffusion (LD) algorithm in eq. (1) which forms the building block for all the algorithms considered in this paper. Intuitively, one should think of (1) as the limit of noisy gradient descent and (2) as the limit of projected noisy gradient descent, both as $\eta \to 0$.

---

**Langevin diffusion.** Let $W_t$ be the standard Brownian motion in $p$-dimensions, and $\beta_t > 0$ be the so called inverse temperature. Then Langevin diffusion is the following stochastic differential equation:

$$d\theta_t = \beta_t \left( -\nabla \mathcal{L}(\theta_t; D) \cdot dt + \frac{\sqrt{2}}{\beta_t} \cdot dW_t \right). \tag{1}$$

**"Projected" Langevin diffusion.** Sometimes, we will only have the Lipschitz guarantee within a constrained set. We can also consider the following "projected" version of LD:

$$d\theta_t = \beta_t \left( -\nabla \mathcal{L}(\theta_t; D) \cdot dt + \frac{\sqrt{2}}{\beta_t} \cdot dW_t + \nu_t L(dt) \right), \forall t \geq 0 : \theta_t \in \mathcal{C}. \tag{2}$$

where $L$ is a measure supported on $\{t : \theta_t \in \partial \mathcal{C}\}$ and $\nu_t$ is an outer unit normal vector at $\theta_t$ for all such $\theta_t$. See e.g. (Bubeck et al., 2018, Section 2.1, 3.1) for a discussion of (2)/verification that a solution exists for convex $\mathcal{C}$ under $M$-smoothness for some finite $M$ (which can be enforced with arbitrarily small perturbations to the loss via convolution).

---

### 1.1 Our Results and Techniques: Conceptual Contributions

**Optimal Excess Population Risk for DP-SCO under $\varepsilon$-DP:** DP-SCO, at this point, is a very well-studied problem in the literature (see references of previous works above). One approach towards obtaining the optimal excess population risk is to first prove an optimal DP-ERM bound, and then use *uniform stability property* (Bousquet & Elisseeff, 2002) (Definition B.14) of the underlying DP algorithm to obtain a population risk guarantee. These two steps indeed provides the optimal bounds of $\Theta\left( \frac{1}{\sqrt{n}} + \frac{\sqrt{p \log(1/\delta)}}{\varepsilon n} \right)$ for convex losses and $\Theta\left( \frac{1}{mn} + \frac{p \log(1/\delta)}{m\varepsilon^2 n^2} \right)$ for $m$-strongly convex losses under $(\varepsilon, \delta)$-DP via variants of DP-SGD (Bassily et al., 2020). One crucial aspect of these bounds is that they reach the non-private optimal SCO bounds as $\varepsilon \to \infty$. In our work, we obtain $\Theta\left( \frac{1}{\sqrt{n}} + \frac{p}{\varepsilon n} \right)$ for convex losses and $\Theta\left( \frac{1}{mn} + \frac{p^2 \log n}{m\varepsilon^2 n^2} \right)$ for $m$-strongly convex losses, under $\varepsilon$-DP. Analogous to the $(\varepsilon, \delta)$-DP setting, these bounds reach the non-private optimal as $\varepsilon \to \infty$. As mentioned earlier, such bounds for the $\varepsilon$-DP setting was unknown prior to this work. Our optimal

| | Assumption | $\varepsilon$-DP | | $(\varepsilon,\delta)$-DP | |
| | | Excess Risk | Time | Excess Risk | Time |
| --- | --- | --- | --- | --- | --- |
| ERM | Convex | $\frac{Lp}{\varepsilon n}$ | $\infty$ | $\frac{L\sqrt{p\log(1/\delta)}}{\varepsilon n}$ | $\frac{1}{p}$ |
| | $m$-SC | $\frac{L^2(p^2+p\log n)}{m\varepsilon^2 n^2}$ | $\infty$ | $\frac{L^2 p\log(1/\delta)}{m\varepsilon^2 n^2}\log^2(p\varepsilon n)$ | $\frac{L^2\log(1/\delta)\log^4(p\varepsilon n)}{m^2\varepsilon^2 n^2}$ |
| | Non-convex | $\frac{Lp}{\varepsilon n}\log\left(\frac{\varepsilon n}{p}\right)$ | $\infty$ | $\frac{Lp}{\varepsilon n}\log\left(\frac{\varepsilon n}{p}\right)$ | $\infty$ |
| SCO | Convex | $\frac{L}{\sqrt{n}}+\frac{Lp}{\varepsilon n}$ | $\infty$ | $\frac{L}{\sqrt{n}}+\frac{L\sqrt{p\log(1/\delta)}}{\varepsilon n}$ | $\min\left\{\frac{1}{p},\frac{\log(1/\delta)}{\varepsilon^2 n}\right\}$ |
| | $m$-SC | $\frac{L^2}{mn}+\frac{L^2 p^2\log n}{m\varepsilon^2 n^2}$ | $\infty$ | $\frac{L^2}{mn}+\frac{L^2 p\log(1/\delta)\log^2(p\varepsilon n)}{m\varepsilon^2 n^2}$ | $\frac{L^2\log(1/\delta)\log^4(p\varepsilon n)}{m^2\varepsilon^2 n^2}$ |

Table 1: Summary of our upper bounds for DP-ERM and DP-SCO. The bounds marked in blue were not known even via different algorithms. Every bound is tight up to polylog $(p,\varepsilon,n)$ factors. Convex: Class of convex bounded Lipschitz losses, $m$-SC: Convex with $\nabla^2\ell(\theta;\cdot)\succcurlyeq m\mathbb{I}$. Non-convex: Class of losses with $\|\nabla\ell(\theta;\cdot)\|_2\leq L$. Here, time ($T$) refers to the length till which we run the Langevin Diffusion algorithm; for the $\varepsilon$-DP results, $T=\infty$ means we use the stationary distribution of the Langevin diffusion. We set the diameter of the constraint set $\|\mathcal{C}\|_2=1$.

DP-SCO bound is obtained by proving a dimension independent uniform stability guarantee for the standard exponential mechanism on the loss function $\mathcal{L}(\theta;D)+\frac{m}{2}\|\theta\|_2^2$. The translation to DP-SCO guarantee is immediate (Bousquet & Elisseeff, 2002) by appealing to the DP-ERM guarantee for exponential mechanism for such score functions, and combining with the uniform stability guarantee.

To show dimension-independent uniform stability of the exponential mechanism on the regularized loss $\mathcal{L}(\theta;D)+\frac{m}{2}\|\theta\|_2^2$, we view the exponential mechanism as the limiting distribution of LD, and provide a time independent $O\left(\frac{L^2}{mn}\right)$-uniform stability of LD. Here, $L$ is the $\ell_2$-Lipschitz parameter of the loss function $\ell(\theta;\cdot)$. Equipped with this bound, we can easily combine with the DP-ERM bound to obtain the excess population risk bound we intended to achieve. We believe the above proof technique can be of independent interest, as it allows one to reduce showing the exponential mechanism is uniformly stable to showing gradient descent has *time-independent uniform stability*, which is a more well-understood problem.

**Unification of DP-ERM and DP-SCO via LD:** In this work we show that LD (Langevin, 1908; Lemons & Gythiel, 1997), defined in (1) and (2), interpolates between optimal known utility bounds for both DP-ERM and DP-SCO, and both under $\varepsilon$- and $(\varepsilon,\delta)$-DP. Based on the inverse temperature, $\beta_t$, and the length for which the diffusion process is run, $T$, Langevin diffusion not only recovers the best known bounds in all the settings mentioned above, but also provides new previously not known results. For example, it recovers the optimal excess population risk bound for convex and strongly convex DP-SCO under $\varepsilon$-DP improving on the previous best known bounds of Asi et al. (2021b). A summary of the results we achieve in this paper are given in Table 1.

While our algorithm is purely information theoretic, it is worth highlighting that it was not clear whether such a universal object that achieves optimal excess risk for both $\varepsilon$- and $(\varepsilon,\delta)$-DP even exists. As we discuss later, the inverse temperature settings of LD that gives rise to the best algorithms for $\varepsilon$-DP and $(\varepsilon,\delta)$-DP are off by roughly a factor of $\sqrt{p}$ ($p$ being the number of model parameters or the dimension of the parameter space). Hence, under constant $\varepsilon$ regime for $(\varepsilon,\delta)$-DP, we analyze LD far before it has converged to a stationary distribution.

**Two-phase analysis of DP-Langevin diffusion:** In the process of demonstrating the universality of LD, we discovered two clear phases in the diffusion process, which enable us to obtain either $(\varepsilon, \delta)$-DP or $\varepsilon$-DP results. For the purposes of brevity, it is easiest to explain this in the context of Lipschitz convex losses. We provide details for other losses in the remainder of the paper. As evident from Table 1 to obtain the DP-ERM and DP-SCO bounds in the $(\varepsilon, \delta)$-DP setting, the LD runs for $T \approx \frac{1}{p}$, whereas, in the $\varepsilon$-DP setting, it needs to run until it convergences to a stationary distribution. It is easy to show that, when $T \leq \frac{1}{p}$, the diffusion process cannot converge to a stationary distribution under reasonable choice of inverse temperature required to ensure DP. This follows from the fact that, within the mentioned time period, for certain loss functions and constraint sets, with high probability, LD does not escape a ball of diameter $c \|\mathcal{C}\|_2$ for $c < 1$ which contains $c^{\Omega(p)}$ of the probability mass of the stationary distribution. However, LD is still able to obtain the desired ERM bounds in this time because the desired ERM bounds are satisfied by points at distance $\Omega(\sqrt{p}/\varepsilon n)$ from the empirical minimizer (see Appendix H for a more detailed discussion). In the $\varepsilon$-DP case, however, we run LD till it has converged to the stationary distribution, which is the exponential mechanism (see Appendix D for a more detailed discussion).

As a result, the utility analysis for these phases are very different. In the $(\varepsilon, \delta)$-DP case, we analyze the algorithm as a noisy gradient flow and use tools from optimization theory (Wilson et al., 2021), whereas, in the $\varepsilon$-DP setting, we analyze the utility in terms of the stationary distribution that the diffusion process converge to, i.e., the Gibbs distribution. Following this viewpoint, if one studies DP-SGD (Bassily et al., 2014) (which is a typical optimization algorithm in the $(\varepsilon, \delta)$-DP case) as a discretization of the LD process, one can observe that under optimal parameter settings it does not converge to anywhere near the stationary distribution (see Appendix H).

## 1.2 OUR RESULTS AND TECHNIQUES: TECHNICAL CONTRIBUTIONS

In this section we give an overview of our technical results. Due to space constraints, we only give formal statements for our most novel results and high-level descriptions for remaining results.

**Rényi divergence bounds for LD (Appendix C and Appendix I):** We cannot use standard composition theorems of DP (Dwork & Roth, 2014) because the underlying algorithm is a continuous time process. One main technical contribution of this work is to quantify the Rényi divergence between two LD processes when run on neighboring data sets:

**Lemma 1.1** (Simplified version of Lemma C.1). *Let $\Theta_{[0,T]}$ be the distribution of the trajectory $\{\theta_t\}_{t \in [0,T]}$ in (2). Suppose we have that $\|\nabla \mathcal{L}(\theta; D) - \nabla \mathcal{L}(\theta; D')\|_2 \leq \Delta$ for all $\theta$. Then for all $\alpha \geq 1$:*

$$R_\alpha(\Theta_{[0,T]}, \Theta'_{[0,T]}) \leq \frac{\alpha \Delta^2}{4} \int_0^T \beta_t^2 dt.$$

The idea behind the above lemma is to define an infinite sequence of pairs of DP-SGD runs on $D, D'$ with decreasing step sizes, such that (i) there is a fixed Rényi divergence bound that holds for all pairs in the sequence and (ii) the trajectory of (2) is the limit of the sequence. We then conclude using Fatou's lemma. This result forms the foundation of the privacy analysis in the rest of the paper. A similar result was also provided in Chourasia et al. (2021). Our result is stronger in that it proves a divergence bound between the entire histories $\{\theta_t\}_{0 \leq t \leq T}$ rather than just the last iterate $\theta_T$, which enables us to output weighted averages of $\theta_t$ privately. Furthermore, it is proven using only tools from the differential privacy literature and Fatou's lemma, providing an arguably much simpler proof. Additionally, for the special case of strongly convex and smooth loss functions, we leverage techniques from Vempala & Wibisono (2019) and Ganesh & Talwar (2020) to show a Rényi-divergence bound (Lemma I.1) that approaches the divergence between the stationary distributions of the LD processes, i.e., the privacy guarantee of the exponential mechanism. Since this bound is not needed for our DP-ERM or DP-SCO results, we defer it to Appendix I.

**LD as exponential mechanism, DP-ERM under $\varepsilon$-DP (Appendix D):** The observation made in privacy analysis of Langevin diffusion (LD) allows us to derive the results on private optimization using exponential mechanism (McSherry & Talwar, 2007). One technical challenge that we need to overcome while mapping LD to exponential mechanism is the issue with optimizing within the constraint set $\mathcal{C}$. We do this via a result of Tanaka (1979), stated in Lemma D.2. This, in turn, readily gives us optimal empirical risk for $\ell_2$-Lipschitz convex functions (described in Appendix D) using the utility analysis in Bassily et al. (2014) (see Theorem D.1). To get optimal empirical risk for strongly convex Lipschitz functions requires an algorithmic improvement. Recall the algorithm in Bassily et al. (2014) for strongly convex losses uses a two step process: output perturbation (Chaudhuri et al., 2011), and then a one shot exponential mechanism. In Algorithm 3, we propose an iterated exponential mechanism which can be of independent interest. This algorithm allows us to define an algorithm purely based on an exponential mechanism over the loss function $\mathcal{L}(\theta; D)$. The idea is to iteratively run the exponential mechanism on a sequence of constraint sets ($\mathcal{C} = \mathcal{C}_0 \supseteq \mathcal{C}_1 \supseteq \ldots \supseteq \mathcal{C}_k$), where $\mathcal{C}_{i+1} = B(\theta_i, r_i) \cap \mathcal{C}_i$, $\theta_i$ being the output of the $\varepsilon_i$-DP exponential mechanism on $\mathcal{C}_i$. For appropriate choices of $\varepsilon_i, r_i$ we get the following ERM bound for $\theta_k$ that is the output of the exponential mechanism on the final constraint set:

**Theorem 1.2** (Simplified version of Theorem D.3). *Algorithm 3 is $\varepsilon$-DP and for m-strongly convex losses, outputs $\theta_k$ such that*

$$\mathbb{E}\left[\mathsf{Risk}_{\mathsf{ERM}}(\theta_k)\right] = O\left(\frac{L^2(p^2 + p\log n)}{m\varepsilon^2 n^2}\right).$$

This is a mild improvement on the excess empirical risk over Bassily et al. (2014), who achieved $O(L^2 p^2 \log n / m\varepsilon^2 n^2)$. For non-convex functions, the results of Bassily et al. (2014) either assume a small ball of radius $r$ is contained in the constraint set $\mathcal{C}$, with utility depending on $\log(\|\mathcal{C}\|_2 / r)$, or use the discrete exponential mechanism on a ball-covering of $\mathcal{C}$. One of our technical contributions is to show that the *continuous* exponential mechanism achieves the optimal excess loss without the small ball assumption (Theorem D.5). Our algorithm is (arguably) more flexible than the ball-covering algorithm in Bassily et al. (2014), since for certain classes of non-convex losses, one can still approximately sample from the continuous exponential mechanism efficiently (e.g., if the stationary distribution satisfies a Poincaré inequality or isoperimetry (Chewi et al., 2021)). Table 1 summarizes these results.

**LD as noisy gradient flow, and DP-ERM under $(\varepsilon, \delta)$-DP (Appendix F.1):** The view of LD we took for the $\varepsilon$-DP case was when the diffusion has converged to a stationary distribution. We also study LD in the setting when it is far from convergence. In fact, we argue in Appendix H that, under the settings we operate with, the algorithm's convergence to the stationary distribution is not much better than a point distribution[2]. We present two results under $(\varepsilon, \delta)$-DP: i) LD achieves optimal excess empirical risk bounds for convex losses (Theorem F.1), and ii) LD achieves optimal excess empirical risk bounds for strongly convex losses (Theorem F.2). The optimality of these algorithms follow from the standard lower bounds in Bassily et al. (2014) and privacy guarantee follows from the privacy accounting machinery described above (see Table 1 for a summary of the bounds).

Our utility bound holds for the last model $\theta_T$ output by LD. On the technical side, for the convex case, we derive a continuous analogue of the excess empirical risk bound for the *last iterate* of noisy SGD in Shamir & Zhang (2013). One of the technical challenges in extending Shamir & Zhang (2013) to continuous case is in finding the best potential function to bound the loss $\mathcal{L}(\theta_T; D)$.

---

[2]There is a setting of parameters obtaining the optimal (asymptotic) excess empirical loss, loss function $\mathcal{L}$, and constraint set $\mathcal{C}$ such that the total variation (1-Wassertein, resp.) distance to the stationary distribution can be as large as $1 - o(1)$ ($\Omega(\|\mathcal{C}\|_2)$, resp.).

Unfortunately, as Shamir & Zhang (2013) also mentions, a direct comparison to $\min\limits_{\theta \in \mathcal{C}} \mathcal{L}(\theta; D)$ does not result in the optimal error. Shamir & Zhang (2013) get around this by bounding the error in terms of the tail averaged loss for a small number of iterates, i.e., $S_k = \frac{1}{k+1} \sum\limits_{t=T-k}^{T} \mathcal{L}(\theta_t; D)$. They then use a recursive argument to compare $S_k$ with $S_{k-1}, S_{k-2}, \cdots$, to finally relate to the required error bound. This approach fails in the continuous setting because there is no concept of a time step. In order to get our bound, we extend the operator $S_k$ to operate over small discretized windows of width $\gamma$ and then use Fatou's lemma to take $\lim\limits_{\gamma \to 0}$ to obtain the continuous analogue. We believe this technique could be of independent interest.

The case for strongly convex loss function is more challenging as the usual choice of potential/Lyupanov[3] function, $\|\theta - \theta_t\|_2^2$, does not immediately provide the right rate of convergence. We pick $e^{mB_t}\|\theta_t - \theta^*\|_2^2$ (based on Wilson (2018)) as the Lyupanov function instead of $\|\theta_t - \theta^*\|_2^2$. (Here, $B_t$ is some fixed function of $t$.) The choice of this Lyupanov function now allows us to get an optimal excess empirical risk guarantee for $\theta^{\mathsf{priv}} = \frac{1}{e^{mB_T}-1} \int_0^T \theta_t d e^{mB_t}$.

**Uniform stability of LD, and unifying convex DP-SCO (Appendices E and F.2):** We finally show the empirical risk to population risk transformation by showing uniform stability bounds for both $\varepsilon$-DP and $(\varepsilon, \delta)$-DP. For the $\varepsilon$-DP setting, standard transformation from empirical risk to population risk in Bassily et al. (2014) (either via $\ell_2$-regularization, or the stability property of DP itself) leads to a bound sub-optimal by a $\sqrt{p}$-factor. We improve this and get an optimal bound as follows:

**Theorem 1.3** (Simplified version of Theorem E.4). *Let $\theta^{\mathsf{priv}}$ be the output of the exponential mechanism when run on the regularized loss $\mathcal{L}_m(\theta; D) := \mathcal{L}(\theta; D) + \frac{m}{2}\|\theta\|_2^2$. For an appropriate choice of $m$ we have:*

$$\mathbb{E}_{\theta^{\mathsf{priv}}}[\mathsf{Risk}_{\mathsf{SCO}}(\theta^{\mathsf{priv}})] = O\left(\frac{Lp\|\mathcal{C}\|_2}{\varepsilon n} + \frac{L\|\mathcal{C}\|_2}{\sqrt{n}}\right).$$

The above bound is obtained by showing a *dimension-independent uniform stability result for the exponential mechanism on strongly convex losses by viewing the exponential mechanism as the limit as as $\eta \to 0, T \to \infty$ of gradient descent, which has a dimension and time-independent uniform stability bound for strongly convex losses* (see Corollary E.3). While Raginsky et al. (2017) as well as extensions of results in Bassily et al. (2014) give dimension-dependent uniform stability bounds for the exponential mechanism, a dimension-independent bound was not known prior to this work. A dimension-independent uniform stability bound was also proven independently of us in the contemporary work of Gopi et al. (2022) (Theorem 6.10), albeit using a different proof. We discuss it in more detail in Appendix A. Given uniform stability of the exponential mechanism on strongly convex losses, we apply it in the convex case by adding a regularizer to the loss function. In the strongly convex case, where we use the iterated exponential mechanism, we show that the population and empirical minimizers of a strongly convex loss function are close to each other with high probability. Given this claim and the uniform stability bound, only a slight modification of our DP-ERM analysis is needed to get the desired DP-SCO bound informally stated as below:

**Theorem 1.4** (Simplified version of Theorem E.6). *Let $\theta_k$ be the output of Algorithm 3 with slight modification to the algorithm's parameters. Then it is $\varepsilon$-DP and for m-strongly convex losses satisfies*

$$\mathbb{E}\left[\mathsf{Risk}_{\mathsf{SCO}}(\theta_k)\right] = O\left(\frac{L^2 p^2 \log n}{m\varepsilon^2 n^2} + \frac{L^2}{mn}\right).$$

---

[3]A Lyapunov function maps scalar or vector variables to real numbers ($\mathbb{R}^p \to \mathbb{R}$) and decreases along the solution trajectory of an SDE. They are primarily used in ordinary differential equation to prove stability and in continuous optimization to prove convergence.

We also prove uniform stability of LD in the $(\varepsilon, \delta)$-DP setting via analyzing noisy gradient descent when learning rate tends to zero. The optimality of the results above follows from standard arguments (see Appendix C in Bassily et al. (2019)[4]).

**Lower-bound for non-convex functions (Appendix G):** In the case of convex loss functions, it is known that we can improve the excess error by $\sqrt{p}$ factor by going from $\varepsilon$-DP to $(\varepsilon, \delta)$-DP. However, the excess empirical risk of our algorithm for non-convex loss function under $(\varepsilon, \delta)$-DP is the same as that under $\varepsilon$-DP. We finally show that it is not an artifact of our algorithm or analysis, rather, in general, it is not possible to get an improvement by going from $\varepsilon$-DP to $(\varepsilon, \delta)$-DP for non-convex loss functions:

**Theorem 1.5** (Simplified version of Theorem G.1)**.** *There exists a dataset $D = \{d_1, \cdots, d_n\}$ and 1-Lipschitz non-convex function $\mathcal{L}$ such that for every $p \in \mathbb{N}$, there is no $(\varepsilon, \delta)$-differentially private algorithm $\mathcal{A}$ that outputs $\theta^{\mathsf{priv}}$ such that $\mathsf{Risk}_{\mathsf{ERM}}(\theta^{\mathsf{priv}}) \in o\left(p \log\left(1/\delta\right)/\left(n\varepsilon\right)\right).$*

The lower bound uses a reduction to the top-$k$-selection problem defined over the universe of size $s$. In particular, we define a packing over the $p$-dimensional Euclidean ball such that there is an bijective mapping between the centers of the packing and $[s]$. Then we define a non-convex function such that the function attains the minimum at the center that corresponds to the coordinate $j \in [s]$ with maximum frequency. Since the size of the $\alpha$-net is $\approx 1/\alpha^p$ and there is a bijective mapping, this gives the desired lower bound using Steinke & Ullman (2017).

## 1.3 RELATED WORKS

**Comparison to Gopi et al. (2022):** In a *concurrent, independent, and complementary* work[5] on convex losses, Gopi et al. (2022) showed that the stationary distribution of a Metropolis-Hastings style process provides the optimal algorithm both for DP-SCO and DP-ERM under $(\varepsilon, \delta)$-DP for $\ell_2$-Lipschitz and convex losses. Their results immediately imply a single algorithm that spans across $\varepsilon$ and $(\varepsilon, \delta)$-DP for DP-ERM. In comparison, our work captures a much larger spectrum of unification, i.e., DP-ERM and DP-SCO, under $\varepsilon$ and $(\varepsilon, \delta)$-DP, for $\ell_2$-Lipschitz losses with/without strong convexity. Furthermore, unlike Gopi et al. (2022) our privacy analysis does not rely on convexity. In terms of gradient oracle complexity, Gopi et al. (2022) give oracle efficient algorithms for their construction. While we acknowledge that the oracle complexity of our LD based algorithm is an important research question, we leave it for future work (see Section 1.4), and focus on statistical efficiency only. The DP-SCO result in Gopi et al. (2022) relies on uniform stability property of exponential mechanism, analogous to our work. While their result relies on bounding the Wasserstein distance between two exponential mechanisms run on neighboring data sets, our uniform stability guarantee follows immediately from the uniform stability guarantee of the diffusion process (which in limit matches the exponential mechanism).

**Comparison to Chourasia et al. (2021); Altschuler & Talwar (2022); Ryffel et al. (2022):** Recently, Chourasia et al. (2021) studied discretization of the LD algorithm as DP-(Stochastic) Gradient Langevin Dynamics (DP-SGLD), and Ryffel et al. (2022) extended these results of Chourasia et al. (2021) to the mini-batch setting. They show that under smoothness and strong convexity on the loss function $\mathcal{L}(\theta; D)$, the privacy cost of DP-SGLD converges to stationary finite value, even when the number of time steps goes to $\infty$. This in-turn improves the gradient oracle complexity over differentially private stochastic gradient descent, dubbed as DP-SGD, (Algorithm 1). Our analysis of DP-LD improves on the result of Chourasia et al. (2021) by allowing divergence bounds

---

[4]The lower bound in Bassily et al. (2019) is technically for $(\varepsilon, \delta)$-DP, but can be interpreted as that for $\varepsilon$-DP up to slack factors of $\mathsf{polylog}(n)$.

[5]The authors of Gopi et al. (2022) also formally acknowledged the independence claim.

between the entire histories $\{\theta_t\}_{0 \leq t \leq T}$ rather than just the last iterate $\theta_T$, which enables us to output weighted averages of $\theta_t$ privately, which is necessary for some of our $(\varepsilon, \delta)$-DP guarantees. Altschuler & Talwar (2022) in a follow-up work to ours, removes the requirement of strong convexity in Chourasia et al. (2021) and provides an analogous bound only for the last iterate $\theta_T$ using a different technique based on optimal transport. In general, these results are orthogonal to us since they do not seek to either unify the existing algorithms or provide tighter utility/privacy trade-offs via the Langevin dynamics/diffusion view point.

**Comparison to Asi et al. (2021b):** Asi et al. (2021b) showed DP-SCO bounds under a general condition called *$\kappa$-growth* via a custom localization based algorithm. A corollary of their results is that under $\varepsilon$-DP one can obtain a excess population risk of $O\left(\frac{\log^{3/2} n}{\sqrt{n}} + \frac{p \log n}{\varepsilon n}\right)$ for Lipschitz convex losses, and $O\left(\frac{\log^3 n}{mn} + \frac{p^2 \log^2 n}{m\varepsilon^2 n^2}\right)$ for Lipschitz and $m$-strongly convex losses under $\varepsilon$-DP. Notice that the bounds do not reach the non-private optimal bounds of $O\left(\frac{1}{\sqrt{n}}\right)$ and $O\left(\frac{1}{mn}\right)$, respectively as $\varepsilon \to \infty$. Our bounds on the other hand have the property of matching the non-private optimal bounds as $\varepsilon \to \infty$. To the best of our knowledge, the polylog(n) dependence in the non-private error bounds of Asi et al. (2021b) is not a slack in the analysis, but is unavoidable for their algorithm. We note that Asi et al. (2021b) has the advantage of requiring weaker primitives (solving an ERM problem instead of the exponential mechanism) and thus being easier to implement. However, our improvements are not solely due to using stronger primitives; e.g., our uniform stability bound is a generalization of a uniform stability bound implicitly used in their paper.

We provide a thorough comparison to other related works in Appendix A.

## 1.4 FUTURE DIRECTIONS

While we know that most of the DP-ERM and DP-SCO bounds are optimal, our understanding of the optimal rates of convergence (in terms of gradient oracle complexity (Bubeck, 2015)) to these error bounds is far from being complete. For example, Kuru et al. (2020) shows DP algorithms with accelerated oracle complexity for strongly convex and smooth losses; can we obtain optimal DP-SCO/DP-ERM rates with accelerated oracle complexity without strong convexity?

Understanding the trajectory of private optimization has further ramifications, such as an understanding of the natural scope of higher order descent methods under privacy constraints, and phenomena that gradients are heavy tailed and lie in a low dimensional subspace. For example, in the non-private setting, higher order methods can be naturally explained using variational methods that study the trajectory of optimization (Wibisono et al., 2016). For DP, one can study the corresponding stochastic variational methods. Here, we can use differentiation as a linear operator and then use the machinery of operator algebra to understand the necessary conditions for the calculus that allows us to derive stochastic variational methods.

From a practical perspective, these methods can be helpful in understanding whether DP-SGD converges to robust network or not when training deep neural network. Without privacy, we know that there is an implicit bias of gradient descent towards non-robust local minima of non-convex problems even though robust networks exist (Vardi et al., 2022). However, because of stochasticity, DP-SGD behaves as the so called *Ornstein-Uhlenbeck process*. An immediate consequence of this phenomenon is that it activates the second-order Taylor's expansion adding a regularizer-like behavior. This phenomenon does not exist in gradient flow and we conjecture that it might be the critical aspect of DP-SGD that allows convergence to a robust network.

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

# A  OTHER RELATED WORK

**Work on optimization:** Although optimization methods in computer science have been mostly discrete, there is a vast literature that studies optimization from a continuous time dynamical point of view, with the earliest example being the mirror-descent (see the discussion in Nemrivosky and Yudin (Nemirovskij & Yudin, 1983, Chapter 3)) and Polyak's momentum method (Polyak, 1964). In fact, Polyak in his 1964 paper motivated his approach using heavy ball moving in a potential and used this physical intuition to give rigorous proof for quadratic loss. More recent works have also shown a closer connection between gradient flow and gradient descent (Wilson, 2018) and between accelerated methods and second-order ordinary differential equations by taking a variational perspective (Hu & Lessard, 2017; Li et al., 2017; Su et al., 2014; Wibisono et al., 2016; Wilson et al., 2021). This, in the case of the unconstrained case, can be interpreted as a damped non-linear oscillator (Cabot et al., 2009). This observation has led to fruitful works to get an averaging interpretation of accelerated dynamics (Krichene et al., 2015; 2016) and also a cornerstone of the "restarting" heuristic (O'donoghue & Candes, 2015).

The idea of approximating discrete-time stochastic algorithms by continuous time equations can be traced back to the vast literature of stochastic approximation theory. We refer the readers to the excellent monograph by Harold, Kushner and Yin on the same topic (Harold et al., 1997). In optimization theory with machine learning as the motivation, the earliest work to the best of our knowledge that studied the dynamical properties of stochastic gradient descent algorithms are the independent and concurrent works of (Li et al., 2017; Mandt et al., 2017; Vollmer et al., 2016). Li et al. (2017) and Mandt et al. (2017) independently gave the first asymptotic convergence of stochastic gradient descent and momentum method as an approximation to stochastic differential equation while Vollmer et al. (2016) gave a non-asymptotic bound on convergence of the stochastic gradient LD algorithm by using the Poisson equations.

The idea of discretizing stochastic differential equations (and by extension stochastic gradient descents and its variants) can be dated back to the seminal work of Mil'shtejn (1975) who performed an extensive numerical analysis of stochastic differential equations. Since then, several works have studied continuous time gradient descent (Hu et al., 2019; Feng et al., 2017), mirror descent (Mertikopoulos & Staudigl, 2018; Raginsky & Bouvrie, 2012), and stochastic variance reduced gradient LD (Dubey et al., 2016) for Bayesian posterior inference.

Some attempts have been made to even study the non-convex setting, considering dissipative loss functions (Raginsky et al., 2017) as well as acceleration (Krichene & Bartlett, 2017) for the vanishing noise limit. In a recent work, Tzen et al. (Tzen et al., 2018) introduced the concept of meta-stability to study the generalization properties of LD for nonconvex functions. This was further generalized and extended by Erdogdu et al. (Erdogdu et al., 2018).

**Work on sampling:** Central to this has been the use of Langevin dynamics that has played a significant role in sampling algorithms. A large line of recent work (including e.g., Dalalyan (2017); Durmus & Moulines (2016); Cheng & Bartlett (2018); Cheng et al. (2018); Vempala & Wibisono (2019); Ganesh & Talwar (2020); Erdogdu et al. (2021); Chewi et al. (2021)) show that using the Langevin dynamics, under certain assumptions (often strong convexity and smoothness or variants thereof) one can efficiently obtain an approximate sample from the stationary distribution. In particular, the gradient oracle complexity of these results is often linear in the dimension and inverse polynomial in the approximation error. The error metric varies from paper to paper; originally, total variation distance, Wasserstein-distances, and KL-divergences were more commonly studied but starting with Vempala & Wibisono (2019), recent works have focused on Rényi divergence bounds.

**Work on DP and optimization:** The connection between dynamical systems and differential privacy is also not new. Chourasia et al. (2021) and Ryffel et al. (2022) study discretization of the LD algorithm as DP-(Stochastic) Gradient Langevin Dynamics (DP-SGLD). They show that under smoothness and strong convexity on the loss function $\mathcal{L}(\theta; D)$, the privacy cost of DP-SGLD converges to a stationary finite value, even when the number of time steps goes to $\infty$. Wang et al. (2019) used the result by Raginsky et al. (2017) to prove a sub-optimal excess empirical risk of $\widetilde{O}\left(\frac{p \log(1/\delta)}{\varepsilon^2 \log(n)}\right)$ for non-convex loss functions. In a concurrent, and complementary work on convex losses, Gopi et al. (2022) study private optimization and show the universality of exponential mechanisms for both stochastic convex optimization and empirical risk minimization. Their analysis takes the sampling perspective when the diffusion process has completed.

It is probably important to mention that objective perturbation (Chaudhuri et al., 2011; Kifer et al., 2012) can be potentially thought of as a (near) universal algorithm for the problem classes considered in this paper, albeit the following two caveats: i) The instantiation of the algorithm for $\varepsilon$-DP and $(\varepsilon, \delta)$-DP require two different noise models to be drawn from, namely, Gamma distribution, and Normal distribution, and ii) It requires the loss functions $\ell(\theta; \cdot)$ to be twice-continuously differentiable, and $\nabla_\theta^2 \ell(\theta; \cdot)$ to have a near constant rank. As mentioned in the remainder of our paper, Langevin diffusion does not require any such assumptions.[6]

## B    NOTATION AND PRELIMINARIES

In this section, we give a brief exposition of the concepts and results required used in the rest of the paper. In Table 2 we provide a summary of all the notation used in the paper.

**Background on Langevin dynamics.**    One of the important tools in stochastic calculus is Ito's lemma (Itô, 1944). It can be seen as the stochastic calculus counterpart of the chain rule and be derived from Taylor's expansion and noting that the second order does not go to zero under quadratic variation:

**Lemma B.1** (Ito's lemma (Itô, 1944)). *Let $\mathbf{x}_t \in \mathbb{R}^p$ be governed by the Langevin diffusion process $d\mathbf{x}_t = \mu_t \cdot dt + \sigma_t \cdot dW_t$, where $W_t$ is the standard Brownian motion in p-dimensions, $\mu_t \in \mathbb{R}^p$ is the drift, and $\sigma_t^2 \in \mathbb{R}$ is the standard deviation. We have the following for any fixed function $f : \mathbb{R}^p \to \mathbb{R}$:*

$$df(\mathbf{x}_t) = \left( \langle \nabla_{\mathbf{x}=\mathbf{x}_t} f(\mathbf{x}), \mu_t \rangle + \frac{\sigma_t^2}{2} \left( \sum_{i=1}^p \left. \frac{\partial^2 f(\mathbf{x})}{\partial x_i^2} \right|_{\mathbf{x}=\mathbf{x}_t} \right) \right) \cdot dt + \sigma_t \cdot \langle \nabla_{\mathbf{x}=\mathbf{x}_t} f(\mathbf{x}), dW_t \rangle .$$

*Here, $\nabla_{\mathbf{x}=\mathbf{x}_t}$ corresponds to $\left[ \frac{\partial}{\partial x_1}, \ldots, \frac{\partial}{\partial x_p} \right]$ evaluated at $\mathbf{x}_t$.*

**Rényi divergence and differential privacy.**    Rényi divergence is the generalization of KL divergence to higher order and satisfies many useful properties (van Erven & Harremos, 2014). More formally,

**Definition B.2** (Rényi Divergence). *For $0 < \alpha < \infty$, $\alpha \neq 1$ and distributions $P, Q$, such that $supp(P) = supp(Q)$ the $\alpha$-Rényi divergence between $P$ and $Q$ is*

---

[6]In particular, we can always ensure twice differentiability by convolving the loss function with the bump kernel (Kifer et al., 2012), and then make the smoothness parameter finite but arbitrarily large which does not affect the Lipschitzness.

| Notation | |
|---|---|
| $D = \{d_1, \cdots, d_n\}$ | data set |
| $\mathcal{D}$ | data distribution |
| $\tau$ | domain set of data |
| $\mathcal{C} \subset \mathbb{R}^p$ | convex set/parameter space |
| $\ell$ | loss function |
| $\mathcal{L}$ | empirical loss function |
| $\mathrm{Risk}_{\mathrm{ERM}}$ | excess empirical risk |
| $\mathrm{Risk}_{\mathrm{SCO}}$ | excess population risk |
| $m$ | strong convexity parameter |
| $M$ | smoothness parameter |
| $L$ | Lipschitz constant |
| $\theta^{\mathrm{priv}}$ | private model output |
| $\theta^*$ | optimal model |
| $\beta_t$ | inverse temperature |
| $\phi$ | Potential/Lyupanov function |
| $W_t$ | standard Brownian motion |
| $R_\alpha(\cdot, \cdot)$ | Renyi divergence of order $\alpha$ |
| $T$ | continuous time |
| $A \succeq 0$ | $A$ is positive semidefinite |
| $A \succeq B$ | $A - B$ is positive semidefinite |
| $\mathbb{I}_p$ | $p \times p$ identity matrix |

Table 2: Notation Table

$$R_\alpha(P, Q) = \frac{1}{\alpha - 1} \ln \int_{supp(Q)} \frac{P(x)^\alpha}{Q(x)^{\alpha - 1}} dx = \frac{1}{\alpha - 1} \ln \mathbb{E}_{x \sim Q} \left[ \frac{P(x)^\alpha}{Q(x)^\alpha} \right].$$

*The $\alpha$-Rényi divergence for $\alpha = 1$ (resp. $\infty$) is defined by taking the limit of $R_\alpha(P, Q)$ as $\alpha$ approaches $1$ (resp. $\infty$) and equals the KL divergence (resp. max divergence).*

We next define differential privacy, our choice of the notion of data privacy. Central to the notion of differential privacy is the definition of underline{adjacent} or underline{neighboring} datasets. Two datasets $D$ and $D'$ are called adjacent if they differ in exactly one data point.

**Definition B.3** (Approximate Differential privacy (Dwork et al., 2006b;a)). *A randomized mechanism $\mathcal{M} : \mathcal{D}^n \to \mathcal{R}$ is said to have $(\varepsilon, \delta)$-differential privacy , or $(\varepsilon, \delta)$-DP for short, if for any adjacent $D, D' \in \mathcal{D}^n$ and measurable subset $S \subset \mathcal{R}$, it holds that*

$$\Pr[\mathcal{M}(D) \in S] \leq e^\varepsilon \Pr[\mathcal{M}(D) \in S] + \delta.$$

*When $\delta = 0$, it is known as pure differential privacy, and we denote it by $\varepsilon$-DP.*

**Definition B.4** (Renyi Differential privacy (Mironov, 2017)). *A randomized mechanism $\mathcal{M} : \mathcal{D}^n \to \mathcal{R}$ is said to have $(\alpha, \varepsilon)$-Rényi differential privacy, or $(\alpha, \varepsilon)$-RDP for short, if for any adjacent $D, D' \in \mathcal{D}^n$ it holds that*

$$R_\alpha(\mathcal{M}(D), \mathcal{M}(D')) \leq \varepsilon.$$

It is easy to see that $\varepsilon$-DP is merely $(\infty, \varepsilon)$-RDP. Similarly, the following fact relates $(\varepsilon, \delta)$-DP to $(\alpha, \varepsilon)$-RDP:

**Fact B.5** ((Mironov, 2017, Proposition 3)). *If $\mathcal{M}$ satisfies $(\alpha, \varepsilon)$-RDP, then $\mathcal{M}$ is $(\varepsilon + \frac{\log 1/\delta}{\alpha - 1}, \delta)$-differentially private for any $0 < \delta < 1$.*

Rényi divergences satisfy a number of other useful properties, which we list here.

**Fact B.6** (Monotonicity (van Erven & Harremos, 2014, Theorem 3)). *For any distributions $P, Q$ and $0 \le \alpha_1 \le \alpha_2$ we have $R_{\alpha_1}(P, Q) \le R_{\alpha_2}(P, Q)$.*

**Fact B.7** (Post-Processing (van Erven & Harremos, 2014, Theorem 9)). *For any sample spaces $\mathcal{X}, \mathcal{Y}$, distributions $P, Q$ over $\mathcal{X}$, and any function $f : \mathcal{X} \to \mathcal{Y}$ we have $R_\alpha(f(P), f(Q)) \le R_\alpha(P, Q)$.*

**Lemma B.8** (Gaussian dichotomy (van Erven & Harremos, 2014, Example 3)). *Let $\mathcal{P} = \mathcal{P}_1 \times \mathcal{P}_2 \times \cdots$ and $\mathcal{Q} = \mathcal{Q}_1 \times \mathcal{Q}_2 \times \cdots$, where $\mathcal{P}_i$ and $\mathcal{Q}_i$ are unit variance Gaussian distributions with mean $\mu_i$ and $\nu_i$, respectively. Then*

$$R_\alpha(\mathcal{P}_i, \mathcal{Q}_i) = \frac{\alpha}{2}(\mu_i - \nu_i)^2,$$

*and by additivity for $\alpha > 0$,*

$$R_\alpha(\mathcal{P}, \mathcal{Q}) = \frac{\alpha}{2} \sum_{i=1}^{\infty} (\mu_i - \nu_i)^2.$$

*As a corollary, we have:*

$$R_\alpha(N(0, \sigma^2 \mathbb{I}_p), N(\mathbf{x}, \sigma^2 \mathbb{I}_p)) \le \frac{\alpha \|\mathbf{x}\|_2^2}{2\sigma^2}.$$

**Fact B.9** (Adaptive Composition Theorem (Mironov, 2017, Proposition 1)). *Let $\mathcal{X}_0, \mathcal{X}_1, \ldots, \mathcal{X}_k$ be arbitrary sample spaces. For each $i \in [k]$, let $f_i, f_i' : \Delta(\mathcal{X}_{i-1}) \to \Delta(\mathcal{X}_i)$ be maps from distributions over $\mathcal{X}_{i-1}$ to distributions over $\mathcal{X}_i$ such that for any distribution $X_{i-1}$ over $\mathcal{X}_{i-1}$, $R_\alpha(f_i(X_{i-1}), f_i'(X_{i-1})) \le \varepsilon_i$. Then, for $F, F' : \Delta(\mathcal{X}_0) \to \Delta(\mathcal{X}_k)$ defined as $F(\cdot) = f_k(f_{k-1}(\ldots f_1(\cdot) \ldots))$ and $F'(\cdot) = f_k'(f_{k-1}'(\ldots f_1'(\cdot) \ldots))$ we have $R_\alpha(F(X_0), F'(X_0)) \le \sum_{i=1}^k \varepsilon_i$ for any $X_0 \in \Delta(\mathcal{X}_0)$.*

**Fact B.10** (Weak Triangle Inequality (Mironov, 2017, Proposition 11)). *For any $\alpha > 1$, $q > 1$ and distributions $\mathcal{P}_1, \mathcal{P}_2, \mathcal{P}_3$ with the same support:*

$$R_\alpha(\mathcal{P}_1, \mathcal{P}_3) \le \frac{\alpha - 1/q}{\alpha - 1} R_{q\alpha}(\mathcal{P}_1, \mathcal{P}_2) + R_{\frac{q\alpha - 1}{q - 1}}(\mathcal{P}_2, \mathcal{P}_3).$$

We discuss two differentially private mechanisms for optimization in this paper. The first one is the exponential mechanism. (McSherry & Talwar, 2007). Given some arbitrary domain $\mathfrak{D}$ and range $\mathfrak{R}$, the exponential mechanism is defined with respect to some loss function, $\ell : \mathfrak{D} \times \mathfrak{R} \to \mathbb{R}$.

**Definition B.11** (Exponential mechanism (McSherry & Talwar, 2007)). *Given a privacy parameter $\varepsilon$, the range $\mathfrak{R}$ and a loss function $\ell : \mathfrak{D} \times \mathfrak{R} \to \mathbb{R}$, the exponential mechanism samples a single element from $\mathfrak{R}$ based on the probability distribution*

$$\pi_D(r) = \frac{e^{-\varepsilon \ell(D, r)/2\Delta_\ell}}{\sum_{r \in \mathfrak{R}} e^{-\varepsilon \ell(D, r)/2\Delta_\ell}}$$

*where $\Delta_\ell$ is the sensitivity of $u$, defined as $\Delta_\ell := \max_{\substack{D \sim D', \\ r \in \mathfrak{R}}} |u(D, r) - u(D', r)|$. If $\mathfrak{R}$ is continuous, we instead sample from the distribution with pdf:*

$$p_D(r) = \frac{e^{-\varepsilon \ell(D, r)/2\Delta_\ell}}{\int_{r \in \mathfrak{R}} e^{-\varepsilon \ell(D, r)/2\Delta_\ell} dr}.$$

---

**Algorithm 1** Differentially private stochastic gradient descent (DP-SGD) (Bassily et al., 2014)

---

**Require:** Data set $D = \{d_1, \cdots, d_n\}$, loss function: $\ell : \mathcal{C} \times \mathcal{D} \to \mathbb{R}$, gradient $\ell_2$-norm bound: $L$,
  constraint set: $\mathcal{C} \subseteq \mathbf{R}^p$, number of iterations: $T$, noise variance: $\sigma^2$, learning rate: $\eta$.
1: Choose any point $\theta_0 \in \mathcal{C}$.
2: **for** $t = 0, \ldots, T-1$ **do**
3:    Sample $d_t$ uniformly at random from $D$ with replacement.
4:    $\nabla \theta^{\mathsf{priv}} \leftarrow \nabla \ell(\theta_t; d_t) + \mathcal{N}\left(0, \sigma^2 \mathbb{I}\right)$, where $\sigma^2 = \frac{8TL^2 \log(1/\delta)}{\varepsilon^2}$.
5:    $\theta_{t+1} \leftarrow \Pi_{\mathcal{C}}\left(\theta_t - \eta \cdot \nabla \theta^{\mathsf{priv}}\right)$, where $\Pi_{\mathcal{C}}(\mathbf{v}) \leftarrow \arg\min_{\theta \in \mathcal{C}} \|\mathbf{v} - \theta\|_2$.
6: **end for**
7: **return** $\theta_t$.

---

The second algorithm that we discuss is the stochastic gradient descent used in Bassily et al. (2014) and presented in Algorithm 1. The algorithm can be seen as noisy stochastic variant of the classic gradient descent algorithm, where stochasticity comes from two sources in every iteration: sampling of $d_t$ and explicit noise addition to the gradient before the descent stage.

We use the result by Steinke & Ullman (2017) for our lower bound proof. We use their equivalent result for empirical mean (see equation (2) in Steinke & Ullman (2017)) and for privacy parameters $(\varepsilon, \delta)$ using a standard reduction (Bun et al., 2018; Steinke & Ullman, 2015)[7]:

**Theorem B.12.** *Fix* $n, s, k \in \mathbb{N}$. *Set* $\beta = 1 + \frac{1}{2} \log\left(\frac{s}{8 \max\{2k, 28\}}\right)$. *Let* $P^1, \cdots, P^s \sim \mathsf{Beta}(\beta, \beta)$ *and let* $X := \{\mathbf{x}_1, \cdots, \mathbf{x}_n\}$ *be such that* $\mathbf{x}_i \in \{0, 1\}^s$ *for all* $i \in [n]$, $\mathbf{x}_{i,j}$ *is independent (conditioned on P) and* $\mathbf{E}[\mathbf{x}_{i,j}] = P^j$ *for all* $i \in [n]$ *and* $j \in [s]$. *Let* $\mathcal{M} : (\{0, 1\}^s)^n \to \{0, 1\}^d$ *be* $(1, \frac{1}{ns})$-*differentially private. Suppose* $\|\mathcal{M}(x)\|_1 = \|\mathcal{M}(x)\|_2^2 = k$ *for all X with probability* 1 *and*

$$\mathop{\mathbf{E}}_{\mathcal{M}}\left[\frac{1}{n}\sum_{i=1}^{n}\sum_{\substack{j \in [s] \\ \mathcal{M}(x)^j = 1}} \mathbf{x}_{i,j}\right] \geq \frac{1}{n}\max_{\substack{S \subset [d] \\ |S| = k}}\sum_{i=1}^{n}\sum_{u \in S} \mathbf{x}_{i,u} - \frac{k}{20}. \tag{3}$$

*Then* $n \in \Omega\left(\sqrt{k} \log\left(\frac{s}{k}\right)\right)$.

**Results from statistics and machine learning:** We will sometimes use Fatou's lemma in our proofs. The form we will use is stated here for convenience:

**Lemma B.13** (Fatou's Lemma). *Let* $\{X_i\}$ *be a sequence of random variables such that there is some constant c such that for all i,* $\mathbf{Pr}[X_i \geq c] = 1$. *Then:*

$$\mathbb{E}\left[\liminf_{i \to \infty} X_i\right] \leq \liminf_{i \to \infty} \mathbb{E}\left[X_i\right].$$

For our SCO bounds, we will use uniform stability. Uniform stability of a learning algorithm is a notion of algorithmic stability introduced to derive high-probability bounds on the generalization error. Formally, it is defined as follows:

---

[7]Steinke & Ullman (2017) present their result in the terms of population mean and privacy parameters $(1, \frac{1}{ns})$.

**Definition B.14** (Uniform stability (Bousquet & Elisseeff, 2002)). *A mechanism $\mathcal{M}$ is $\mu(n)$-uniformly stable with respect to $\ell$ if for any pair of databases $D, D'$ of size $n$ differing in at most one individual:*

$$\sup_{d \in \tau} \left[ \mathbb{E}_{\mathcal{M}} \left[ \ell(\mathcal{M}(D), d) \right] - \mathbb{E}_{\mathcal{M}} \left[ \ell(\mathcal{M}(D'), d) \right] \right] \leq \mu(n).$$

In this paper, we will need the following result.

**Lemma B.15** (Bousquet & Elisseeff (2002)). *Suppose $\mathcal{M}$ is $\mu(n)$-uniformly stable. Then:*

$$\mathbb{E}_{D \sim \mathcal{D}^n, \mathcal{M}} [\mathsf{Risk_{SCO}}(\mathcal{M}(D))] \leq \mathbb{E}_{D \sim \mathcal{D}^n, \mathcal{M}} [\mathsf{Risk_{ERM}}(\mathcal{M}(D))] + \mu(n).$$

## C    RÉNYI DIVERGENCE BOUND FOR LANGEVIN DIFFUSION (LD)

This section is devoted to proving the divergence bounds between two LD processses when run on neighboring datasets. It forms the basis of the privacy analysis in the rest of the paper.

**Lemma C.1.** *Let $\theta_0, \theta'_0$ have the same distribution $\Theta_0$, $\theta_T$ be the solution to (2) given $\theta_0$ and database $D$, and let $\theta'_T$ be the solution to (2) given $\theta_0$ and database $D'$, such that $D \sim D'$. Let $\Theta_{[0,T]}$ be the distribution of the trajectory $\{\theta_t\}_{t \in [0,T]}$. Suppose we have that $\|\nabla \mathcal{L}(\theta; D) - \nabla \mathcal{L}(\theta; D')\|_2 \leq \Delta$ for all $\theta$. Then for all $\alpha \geq 1$:*

$$R_\alpha(\Theta_{[0,T]}, \Theta'_{[0,T]}) \leq \frac{\alpha \Delta^2}{4} \int_0^T \beta_t^2 dt.$$

The idea behind the proof is to use a bound on the divergence between Gaussians and RDP composition to provide a bound on the divergence between the projected noisy gradient descents on datasets $D$ and $D'$. Then, taking the limit as the step size in gradient descent goes to 0 and applying Fatou's lemma (Lemma B.13), we get the bound above.

*Proof.* For ease of presentation, we will show a divergence bound between $\Theta_T$, $\Theta'_T$ which are the distributions of $\theta_t, \theta'_t$, and then describe how to modify the proof to show the same bound between $\Theta_{[0,T]}, \Theta'_{[0,T]}$.

Let $\Psi_{D,m,i}$ be a map from (distributions over) $\mathbb{R}^p$ to (distributions over) $\mathbb{R}^p$ that takes the point $\theta$ to the distribution $\Pi_{\mathcal{C}} \left( N \left( \theta - \left( \int_{(i-1)T/m}^{iT/m} \beta_t dt \right) \nabla \mathcal{L}(\theta; D), 2\frac{T}{m}\mathbb{I} \right) \right)$, where $\Pi_{\mathcal{C}}$ is the $\ell_2$-projection into $\mathcal{C}$. It is well known (see e.g. Lemma B.8) that:

$$R_\alpha(N(0, \sigma^2\mathbb{I}), N(\mathbf{x}, \sigma^2\mathbb{I})) \leq \frac{\alpha \|\mathbf{x}\|_2^2}{2\sigma^2}.$$

So by post-processing (Fact B.7) and the Lipschitzness assumption, $R_\alpha(\Psi_{D,m,i}(\theta), \Psi_{D',m,i}(\theta))$ is bounded by

$$R_\alpha \left( N \left( \theta - \left( \int_{(i-1)T/m}^{iT/m} \beta_t dt \right) \nabla \mathcal{L}(\theta; D), \frac{2T}{m} \mathbb{I} \right), N \left( \theta - \left( \int_{(i-1)T/m}^{iT/m} \beta_t dt \right) \nabla \mathcal{L}(\theta; D'), \frac{2T}{m} \mathbb{I} \right) \right)$$

$$= R_\alpha \left( N \left( \mathbf{0}, \frac{2T}{m} \mathbb{I} \right), N \left( \left( \int_{(i-1)T/m}^{iT/m} \beta_t dt \right) (\nabla \mathcal{L}(\theta; D) - \nabla \mathcal{L}(\theta; D')), 2\frac{T}{m} \mathbb{I} \right) \right)$$

$$\leq \frac{\alpha \Delta^2}{4} \cdot \frac{\left( \int_{(i-1)T/m}^{iT/m} \beta_t dt \right)^2}{T/m}.$$

Let $\Psi_{D,m}$ denote the composition $\Psi_{D,m,m} \circ \Psi_{D,m,m-1} \circ \ldots \circ \Psi_{D,m,1}$. By Fact B.9, we have

$$R_\alpha(\Psi_{D,m}(\Theta_0), \Psi_{D',m}(\Theta_0)) \leq \sum_{i=1}^{m} \max_{\theta} \left\{ R_\alpha(\Psi_{D,m,i}(\theta), \Psi_{D',m,i}(\theta)) \right\}.$$

Plugging in the bound on $R_\alpha(\Psi_{D,m,i}(\theta), \Psi_{D',m,i}(\theta))$, we get

$$R_\alpha(\Psi_{D,m}(\Theta_0), \Psi_{D',m}(\Theta_0)) \leq \frac{\alpha \Delta^2}{4} \cdot \frac{m}{T} \sum_{i=1}^{m} \left( \int_{(i-1)T/m}^{iT/m} \beta_t dt \right)^2$$

Note that $\Theta_T = \lim_{m\to\infty} \Psi_{D,m}(\Theta_0)$, and $\Theta_T' = \lim_{m\to\infty} \Psi_{D',m}(\Theta_0)$. Since $\exp((\alpha - 1)R_\alpha(\mathcal{P}, \mathcal{Q}))$ is a monotone function of $R_\alpha(\mathcal{P}, \mathcal{Q})$ and is the expectation of a positive random variable, by Fatou's lemma we have:

$$R_\alpha(\Theta_T, \Theta_T') \leq \lim_{m\to\infty} R_\alpha(\Psi_{D,m}(\Theta_0), \Psi_{D',m}(\Theta_0))$$

$$\leq \frac{\alpha \Delta^2}{4} \cdot \lim_{m\to\infty} \frac{m}{T} \sum_{i=1}^{m} \left( \int_{(i-1)T/m}^{iT/m} \beta_t dt \right)^2 = \frac{\alpha \Delta^2 \int_0^T \beta_t^2 dt}{4}.$$

This gives the bound on $R_\alpha(\Theta_T, \Theta_T')$. To obtain the same bound for $R_\alpha(\Theta_{[0,T]}, \Theta_{[0,T]}')$, we modify $\Psi_{D,m,i}$ so that instead of receiving $\Theta_{(i-1)T/m}$ and outputting $\Theta_{iT/m}$, it receives the joint distribution $\{\Theta_{jT/m}\}_{0\leq j\leq i-1}$ and outputs $\{\Theta_{jT/m}\}_{0\leq j\leq i}$ by appending the (also jointly distributed) variable

$$\Theta_{iT/m} = \Pi_\mathcal{C} \left( N \left( \theta - \left( \int_{(i-1)T/m}^{iT/m} \beta_t dt \right) \nabla \mathcal{L}(\Theta_{(i-1)T/m}; D), 2\frac{T}{m} \mathbb{I} \right) \right)$$

That is, we update $\Psi_{D,m,i}$ so it outputs the distributions of all iterates seen so far instead of just the distribution of the last iterate; the limiting value of the joint distribution $\{\Theta_{jT/m}\}_{0\leq j\leq i}$ is then $\Theta_{[0,T]}$ according to *eq.* (2), and the same divergence bound holds. $\qquad\square$

# D  LD AS EXPONENTIAL MECHANISM AND DP-ERM UNDER $\varepsilon$-DP

In this section, we study the privacy-utility trade-offs of LD when viewed as variants of exponential mechanism (McSherry & Talwar, 2007). Using this view point we show that LD can achieve the optimal excess empirical risk bounds for $\ell_2$-Lipschitz losses, including non-convex, convex, and strongly convex losses.

## D.1  BOUND FOR CONVEX LOSSES

We revisit the result from Bassily et al. (2014) for convex losses, and quote the result for completeness purposes. Notice that the privacy guarantee in Theorem D.1 *does not rely on convexity*.

---

**Algorithm 2** Exponential mechanism

---

**Input:** Loss function $\mathcal{L}$, constraint set $\mathcal{C} \subset \mathbb{R}^p$ with bounded diameter, Lipschitz constant $L$, number of iterations $k$, privacy parameter $\varepsilon$, data set $D$ of $n$-samples.

1: Sample and **output** a point $\theta^{\mathsf{priv}}$ from the constraint set $\mathcal{C}$ w.p. $\propto \exp\left(-\frac{\varepsilon n}{2L\|\mathcal{C}\|_2} \cdot \mathcal{L}(\theta; D)\right)$.

---

**Theorem D.1.** *Assume each of the individual loss function in $\mathcal{L}(\theta; D)$ is L-Lipschitz within the constraint set $\mathcal{C}$, individual loss function $\ell(\theta; \cdot)$ is convex, and the constraint set $\mathcal{C}$ is convex. Then, Algorithm 2 is $\varepsilon$-differentially private. Furthermore, for $\theta^{\mathsf{priv}}$ as specified in Algorithm 2, over the randomness of the algorithm,*

$$\mathbb{E}_{\theta^{\mathsf{priv}}}[\mathsf{Risk}_{\mathsf{ERM}}(\theta^{\mathsf{priv}})] = O\left(\frac{Lp \cdot \|\mathcal{C}\|_2}{\varepsilon n}\right).$$

**Equivalence of Algorithm 2 and Langevin diffusion:** The following lemma, which is implied by, e.g. (Tanaka, 1979, Theorem 4.1), shows that one can implement Algorithm 1 using only solutions to eq. (2); note that this does not necessarily mean solutions to eq. (2) are efficiently sampleable.

**Lemma D.2.** *Let $\mathcal{L}$ be a M-smooth function for some finite M. Then if $\beta_t = \beta$ for all $t$, then the stationary distribution of (2) has pdf proportional to $\exp(-\beta\mathcal{L}(\theta; D)) \cdot \mathbb{1}(\theta \in \mathcal{C})$, where $\mathbb{1}(\cdot)$ is the indicator function.*

We recall that one can ensure smoothness by convolving $\mathcal{L}$ (appropriately extended to all of $\mathbb{R}^p$) with the Gaussian kernel of finite variance (Feldman et al., 2018, Appendix C). In particular, since we only need $M$ to be finite, we can take the convolution with the Gaussian kernel $\mathcal{N}(\mathbf{0}, \lambda^2\mathbb{I}_p)$ for arbitrarily small $\lambda > 0$, and in turn the result of the convolution is $L/\lambda$-smooth (which is perhaps arbitrarily large but still finite) and differs from $\mathcal{L}$ by an arbitrarily small amount everywhere in $\mathcal{C}$.

## D.2  BOUND FOR STRONGLY CONVEX LOSSES

Our algorithm for strongly convex losses, given as Algorithm 3, is an iterated version of the exponential mechanism. Again, we note that Algorithm 3 can be implemented using only the Langevin Diffusion as a primitive.

**Theorem D.3.** *For any $\varepsilon$, suppose we run Algorithm 3 with $k = 1 + \lceil\log\log(\frac{\varepsilon n}{(p+\log n)})\rceil$, and $\varepsilon_i = \varepsilon/2^{k-i+1}$. Assume each of the individual loss function in $\mathcal{L}(\theta; D)$ is L-Lipschitz within the constraint set $\mathcal{C}_0$. Then Algorithm 3 is $\varepsilon$-differentially private. Additionally, if the loss function $\mathcal{L}(\theta; D)$ is m-strongly convex, and the constraint set $\mathcal{C}_0$ is convex, then over the randomness of the algorithm, the output $\theta_k$ of Algorithm 3 satisfies:*

$$\mathbb{E}[\mathsf{Risk}_{\mathsf{ERM}}(\theta_k)] = O\left(\frac{L^2(p^2 + p\log n)}{m\varepsilon^2 n^2}\right).$$

---

**Algorithm 3** Iterated Exponential Mechanism (ITERATED-EXP-MECHANISM)

---

**Input:** Loss function $\mathcal{L}$, constraint set $\mathcal{C}_0 \subset \mathbb{R}^p$ with bounded diameter, Lipschitz constant $L$, strong convexity parameter $m$, number of iterations $k$, privacy parameter sequence $\{\varepsilon_i\}_{i=1}^k$, and data set $D$ of $n$ samples.

1: **for** $i = 1$ to $k$ **do**
2:      Sample $\theta_i$ from $\mathcal{C}_{i-1}$ with probability proportional to $\exp(-\frac{\varepsilon_i n}{2L\|\mathcal{C}_{i-1}\|_2}\mathcal{L}(\theta; D))$.
3:      $\mathcal{C}_i \leftarrow \left\{ \theta \in \mathcal{C}_{i-1} : \|\theta - \theta_i\|_2 \leq \sqrt{\frac{cL(p+3\log n)\|\mathcal{C}_{i-1}\|_2}{m\varepsilon_i n}} \right\}$.
4: **end for**
5: **return** $\theta_k$

---

The theorem follows by solving a recurrence for $\|\mathcal{C}_i\|_2$ to bound the diameter of the final set $\mathcal{C}_{k-1}$. Then, we show that the minimizer over $\mathcal{C}_0$ is also in $\mathcal{C}_{k-1}$ with high probability, so the analysis of the exponential mechanism gives the theorem. We also note that a similar result is achieved by Bassily et al. (2014). However, we improve the $p^2 \log n$ in their result to $p^2 + p \log n$, and only need the exponential mechanism as a primitive (whereas their algorithm requires computing the minimum of $\mathcal{L}$).

To bound the prove Theorem D.3, we first need the following lemma, which shows that with high probability we choose a series of $\mathcal{C}_i$ that all contain the optimal $\theta$ for $\mathcal{C}_0$.

**Lemma D.4.** *Suppose we sample $\theta$ from the convex constraint set $\mathcal{C} \subset \mathbb{R}^p$ with bounded diameter with probability proportional to $\exp\left(-\frac{\varepsilon n}{2L\|\mathcal{C}\|_2}\mathcal{L}(\theta; D)\right)$, where $\mathcal{L}(\cdot; D)$ is an $m$-strongly convex function. Let $\theta^* = \arg\min_{\theta \in \mathcal{C}} \mathcal{L}(\theta; D)$. Then for any $t \geq 0$ and for some sufficiently large constant $c$ we have*

$$\mathbf{Pr}\left[ \|\theta - \theta^*\|_2 \leq \sqrt{\frac{cL(p+t)\|\mathcal{C}\|_2}{m\varepsilon n}} \right] \geq 1 - 2^{-t}.$$

The lemma follows from a tail bound on the excess loss of the exponential mechanism, and using $m$-strong convexity to translate this into a distance bound. The proof is given below.

*Proof of Lemma D.4.* By e.g. the proof of (Bassily et al., 2014, Theorem III.2), we know that for some sufficiently large constant $c$:

$$\mathbf{Pr}\left[ \mathcal{L}(\theta; D) - \mathcal{L}(\theta^*; D) \leq \frac{cL\|\mathcal{C}\|_2}{2\varepsilon n}(p+t) \right] \geq 1 - 2^{-t}. \tag{4}$$

We now show that the claim holds conditioned on this event. By optimality of $\theta^*$ and convexity of $\mathcal{C}$, we know

$$\langle \nabla\mathcal{L}(\theta^*; D), \theta - \theta^* \rangle \geq 0. \tag{5}$$

So, by $m$-strong convexity, we have

$$\frac{cL\|\mathcal{C}\|_2}{2\varepsilon n}(p+t) \underset{(4)}{\geq} \mathcal{L}(\theta; D) - \mathcal{L}(\theta^*; D) \geq \langle \nabla\mathcal{L}(\theta^*; D), \theta - \theta^* \rangle + \frac{m}{2}\|\theta - \theta^*\|_2^2 \underset{(5)}{\geq} \frac{m}{2}\|\theta - \theta^*\|_2^2.$$

Rearranging gives the claim. □

Given Lemma D.4, we can now prove Theorem D.3.

*Proof of Theorem D.3.* The privacy guarantee is immediate from the privacy guarantee of the exponential mechanism, composition, and the fact that for this choice of $\varepsilon_i, k$, we have $\sum_{i=1}^{k} \varepsilon_i < \varepsilon$.

Setting $t = 3 \log n$ in Lemma D.4, in iteration $i$, letting $\theta_i^* = \arg\min_{\theta \in \mathcal{C}_{i-1}} \mathcal{L}(\theta; D)$, we have that with probability $1 - 2^{-t} = 1 - \frac{1}{n^3}$, $\theta_i^* \in \mathcal{C}_i$, and thus $\theta_i^* = \theta_{i+1}^*$. Then by a union bound, we have that with probability $1 - \frac{k}{n^3} \geq 1 - \frac{\log\log(\varepsilon n)}{n^3}$, $\theta_1^* \in \mathcal{C}_{k-1}$ (equivalently, $\theta_1^* = \theta_2^* = \ldots = \theta_k^*$). When this event fails to happen, our excess loss is at most $L \|\mathcal{C}_0\|_2$, and in turn the contribution of this event failing to hold to the expected excess loss is $O(\frac{L\|\mathcal{C}_0\|_2 \log\log(\varepsilon n)}{n^3})$, which is asymptotically less than our desired excess loss bound. So it suffices to provide the desired expected excess loss bound conditioned on this event. By the analysis of the exponential mechanism, conditioned on this event, we have that

$$\mathbb{E}_{\theta_k}[\mathcal{L}(\theta_k; D)] - \mathcal{L}(\theta_1^*; D) = O\left(\frac{Lp \|\mathcal{C}_{k-1}\|_2}{\varepsilon_k n}\right) = O\left(\frac{Lp \|\mathcal{C}_{k-1}\|_2}{\varepsilon n}\right). \tag{6}$$

Note that $\mathcal{L}(\theta_1^*; D) = \min_{\theta \in \mathcal{C}_0} \mathcal{L}(\theta; D)$ by definition, so it now suffices to bound $\|\mathcal{C}_{k-1}\|_2$ by $O\left(\frac{L(p+\log n)}{m\varepsilon n}\right)$. To do this, we have the recurrence relation:

$$\|\mathcal{C}_i\|_2 \leq 2\sqrt{\frac{cL(p + 3\log n) \|\mathcal{C}_{i-1}\|_2}{m\varepsilon_i n}}.$$

Solving the recurrence relation for $\mathcal{C}_{k-1}$, we get:

$$\begin{aligned}
\|\mathcal{C}_{k-1}\|_2 &\leq \left(\frac{4cL(p + 3\log n)}{mn}\right)^{1-2^{-(k-1)}} \cdot (\|\mathcal{C}_0\|_2)^{2^{-(k-1)}} \cdot \prod_{i=1}^{k-1} \varepsilon_i^{-2^{-(k-i)}} \\
&= \left(\frac{4cL(p + 3\log n)}{m\varepsilon n}\right)^{1-2^{-(k-1)}} \cdot (\|\mathcal{C}_0\|_2)^{2^{-(k-1)}} \cdot \prod_{i=1}^{k-1} (2^{(k-i+1)})^{2^{-(k-i)}}.
\end{aligned} \tag{7}$$

We claim the following:

$$\|\mathcal{C}_0\|_2 \leq \frac{2L}{m}. \tag{8}$$

Let $\theta_{\text{global}}$ be the minimizer of $\mathcal{L}(\theta; D)$ over all of $\mathbb{R}^p$. By triangle inequality, there exists a point $\theta$ in $\mathcal{C}_0$ which is at distance at least $\|\mathcal{C}_0\|_2 / 2$ far from $\theta_{\text{global}}$. By $m$-strong convexity, this implies that the gradient at $\theta$ has $\ell_2$-norm at least $m \|\mathcal{C}_0\|_2 / 2$. Now, by Lipschitzness over $\mathcal{C}_0$, we know that the gradient at $\theta$ has $\ell_2$-norm at most $L$. This gives us eq. (8).

Using eq. (8), we can simplify eq. (7) to

$$\|\mathcal{C}_{k-1}\|_2 \leq \frac{2L}{m} \cdot \left(\frac{2c(p+3\log n)}{\varepsilon n}\right)^{1-2^{-(k-1)}} \cdot \prod_{i=1}^{k-1} (2^{(k-i+1)})^{2^{-(k-i)}}.$$

We have:

$$\log_2\left(\prod_{i=1}^{k-1}(2^{(k-i+1)})^{2^{-(k-i)}}\right) = \sum_{i=1}^{k-1}(k-i+1)2^{-(k-i)} \leq \sum_{j=1}^{\infty}(j+1)2^{-j} = 3.$$

In other words, $\prod_{i=1}^{k-1}(2^{(k-i+1)})^{2^{-(k-i)}}$ is at most 8, regardless of the value of $k$. Now, using the fact that $m^{1/\log m} = O(1)$ is a constant, our final upper bound on $\|\mathcal{C}_{k-1}\|_2$ is:

$$\|\mathcal{C}_{k-1}\|_2 = O\left(\frac{L}{m} \cdot \left(\frac{(p+\log n)}{\varepsilon n}\right)^{1-2^{-(k-1)}}\right) = O\left(\frac{L(p+\log n)}{m\varepsilon n}\right).$$

Plugging in eq. (6) gives us Theorem D.3. □

## D.3 BOUND FOR NON-CONVEX LOSSES

If the loss function $\mathcal{L}$ is non-convex but still $L$-Lipschitz, we can still obtain a comparable error bound for Algorithm 2 as long as the constraint set $\mathcal{C}$ is convex.

**Theorem D.5.** *Assume the constraint set $\mathcal{C} \subset \mathbb{R}^p$ with bounded diameter is convex, and each of the individual loss function in $\mathcal{L}(\theta; D)$ is L-Lipschitz within $\mathcal{C}$. Then for $p \leq \varepsilon n/2$, over the randomness of Algorithm 2, the output $\theta^{\mathsf{priv}}$ satisfies*

$$\mathbb{E}\left[\mathsf{Risk}_{\mathsf{ERM}}(\theta^{\mathsf{priv}})\right] = O\left(\frac{Lp \cdot \|\mathcal{C}\|_2}{\varepsilon n}\log\left(\frac{\varepsilon n}{p}\right)\right).$$

Note that the assumption on $p$ can easily be removed: if $p > \varepsilon n/2$, any $\theta \in \mathcal{C}$ achieves $\mathsf{Risk}_{\mathsf{ERM}}(\theta) \leq L\|\mathcal{C}\|_2 < \frac{2Lp \cdot \|\mathcal{C}\|_2}{\varepsilon n}$ by $L$-Lipschitzness. To prove Theorem D.5, we show there is a "good" subset of $\mathcal{C}$, $\mathcal{C}_{\mathsf{Good}} \subset \mathcal{C}$, with large volume and only containing points with small excess loss. We note that Bassily et al. (2014) also gave an analysis for the continuous exponential mechanism on non-convex losses, although their analysis assumes $\mathcal{C}$ contains an $\ell_2$-ball of radius $r > 0$, which they use to choose $\mathcal{C}_{\mathsf{Good}}$ in their analysis. In turn, their error bound is roughly proportional to $\log(1/r)$. In contrast, by choosing $\mathcal{C}_{\mathsf{Good}}$ more carefully, we remove this dependence on $r$.

*Proof.* By the analysis of the exponential mechanism, for any $\mathcal{G} \subseteq \mathcal{C}$ we have:

$$\mathbb{E}_{\theta^{\mathsf{priv}}}\left[\mathcal{L}(\theta^{\mathsf{priv}};D)\right] - \max_{\theta \in \mathcal{G}}\mathcal{L}(\theta;D) = O\left(\frac{L\|\mathcal{C}\|_2}{\varepsilon n} \cdot \log\left(\frac{\mathsf{Vol}(\mathcal{C})}{\mathsf{Vol}(\mathcal{G})}\right)\right).$$

Let us define $\mathcal{G}$ to be
$$\mathcal{G} := \{\theta^* + r(\theta - \theta^*) : \theta \in \partial\mathcal{C}, 0 \leq r \leq R\},$$
for some $R \leq 1$ we will choose later, where $\theta^*$ is a minimizer of $\mathcal{L}(\theta; D)$ in $\mathcal{C}$, and $\partial\mathcal{C}$ is the boundary of $\mathcal{C}$. By convexity of $\mathcal{C}$, $\mathcal{G}$ is contained in $\mathcal{C}$. Furthermore, $\mathcal{G}$ is simply $\mathcal{C}$ rescaled by

$R$ in all directions around the point $\theta^*$ (i.e., if $\theta^*$ were the origin, $\mathcal{G}$ would simply be $R\mathcal{C}$), so $\frac{\text{Vol}(\mathcal{C})}{\text{Vol}(\mathcal{G})} = (1/R)^p$. So the analysis of the exponential mechanism gives

$$\mathbb{E}_{\theta^{\text{priv}}} \left[ \mathcal{L}(\theta^{\text{priv}}; D) \right] - \max_{\theta \in \mathcal{G}} \mathcal{L}(\theta; D) = O\left( \frac{Lp \|\mathcal{C}\|_2}{\varepsilon n} \cdot \log\left( \frac{1}{R} \right) \right). \tag{9}$$

By $L$-Lipschitzness of $\mathcal{L}(\theta; D)$ and since $\mathcal{G}$ is $\mathcal{C}$ rescaled by $R$, and thus $\max_{\theta \in \mathcal{G}} \|\theta - \theta^*\|_2 \le R \|\mathcal{C}\|_2$, we have:

$$\max_{\theta \in \mathcal{G}} \mathcal{L}(\theta; D) - \mathcal{L}(\theta^*; D) \le RL \|\mathcal{C}\|_2. \tag{10}$$

Combining (9) and (10), we get:

$$\mathbb{E}_{\theta^{\text{priv}}} \left[ \text{Risk}_{\text{ERM}}(\theta^{\text{priv}}) \right] = O\left( L \|\mathcal{C}\|_2 \left( \frac{p}{\varepsilon n} \cdot \log\left( \frac{1}{R} \right) + R \right) \right). \tag{11}$$

The above bound is minimized by choosing $R = \frac{p}{\varepsilon n}$, which is at most 1 by assumption, giving the theorem. $\qquad\square$

## E  UNIFORM STABILITY OF LD AND OPTIMAL DP-SCO UNDER $\varepsilon$-DP

In this section we provide the uniform stability bounds for LD in the setting of $\varepsilon$-DP. These bounds combined with excess empirical risk bounds in Section D provide us optimal excess population risk bounds for convex losses and strongly convex losses.

### E.1  BOUND FOR CONVEX LOSSES

All our algorithms in the $\varepsilon$-DP case are primarily based on the exponential mechanism. Therefore, to get SCO bound, we establish uniform stability of the exponential mechanism on regularized losses. To the best of our knowledge, such uniform stability bounds for exponential mechanism were unknown prior to this work. We first recall the following fact about projected noisy gradient descent on strongly convex (and smooth) losses.

**Lemma E.1.** *Let $f, f'$ be $m$-strongly convex, $M$-smooth functions such that $\|\nabla f(\theta) - \nabla f'(\theta)\|_2 \le \Delta$ for all $\theta \in \mathcal{C}$. Recall that given convex set $\mathcal{C}$, projected noisy gradient descent on $f$ performs the following random update: sample $\xi_t \sim N(\mathbf{0}, \sigma^2 \mathbb{I}_p)$ and compute $\theta_{t+1} = \Pi_{\mathcal{C}} \left( \theta_t - \eta_t \nabla f(\theta_t) + \xi_t \right)$. Here $\Pi_{\mathcal{C}}$ is the Euclidean projection onto $\mathcal{C}$. Let $\{\theta_t\}_t$, $\{\theta'_t\}_t$ be the trajectories given by running projected noisy gradient descent on $f, f'$ respectively starting from the same point $\theta_0$, and suppose $M \le 1/\max_t \eta_t$. Then for any (shared) fixed realization of the noise $\{\xi_t\}_t$, $\|\theta_t - \theta'_t\|_2 \le \frac{\Delta}{m}$.*

Lemma E.1 is implied by e.g. the proof of (Hardt et al., 2016, Theorem 3.9). We have the following corollaries:

**Corollary E.2.** *If $\ell$ is $m$-strongly convex and $L$-Lipschitz, then for any $\beta_t$ such that $\int_a^b \beta_t dt$ is finite if $0 \le a \le b$, and any $t > 0$, outputting $\theta_t$ that is the solution to (2) (given some fixed $\theta_0$ independent of $\mathcal{L}$) is $\frac{2L^2}{mn}$-uniformly stable.*

**Corollary E.3.** *If $\ell$ is convex and $L$-Lipschitz, then running Algorithm 2 on the regularized loss function $\mathcal{L}_m(\theta; D) := \mathcal{L}(\theta; D) + \frac{m}{2} \|\theta\|_2^2$ is $\frac{2L^2}{mn}$-uniformly stable (with respect to the unregularized loss $\mathcal{L}$).*

The idea behind the proofs is that the uniform stability bound implied by Lipschitzness and Lemma E.1 is independent of $t, \eta$, and so it applies to the limiting distribution as $\eta \to 0$ (giving Corollary E.2) and as $t \to \infty$, which by Lemma D.2 is the exponential mechanism (giving Corollary E.3). We believe that this proof demonstrates the power of the unified framework that is the focus of this paper; once we have this framework in mind, we can almost immediately obtain optimal SCO bounds under $\varepsilon$-DP and also improve on Asi et al. (2021b) by polylog $(n)$ factors.

*Proof of Corollary E.2.* Since the regularizer does not depend on the dataset, $D$, by $L$-Lipschitzness we have

$$\left\| \nabla \mathcal{L}(\theta; D) - \nabla \mathcal{L}(\theta; D') \right\|_2 \leq \frac{2\beta L}{n}.$$

Fix some $\eta > 0$ and set $f = \mathcal{L}, \eta_t = \int_{(t-1)\eta}^{t\eta} \beta_t dt$, $\sigma^2 = 2\eta$ in Lemma E.1, and note that the bound in Lemma E.1 does not depend on $\sigma^2$. We will assume $\mathcal{L}$ is $M$-smooth for some finite $M$, since per the discussion in Section D, we can easily replace $\mathcal{L}$ with a smoothed version of $\mathcal{L}$ that differs from $\mathcal{L}$ by an arbitrarily small amount. Lemma E.1 then gives that, if $\theta_t, \theta_t'$ are the result of running projected noisy gradient descent on $\mathcal{L}$ using $D, D'$ respectively (from the same initial $\theta_0$), then we have $\left\| \theta_t - \theta_t' \right\|_2 \leq \frac{2L}{mn}$ for any fixed realization of the noise $\{\xi_t\}_t$ as long as the smoothness assumption is satisfied. Taking the limit as $\eta$ goes to 0, the trajectory of this projected noisy gradient descent approaches the solution to (2). Furthermore, taking this limit the smoothness assumption in Lemma E.1 is trivially satisfied as all $\eta_t$ go to 0. Therefore, by Fatou's lemma (Lemma B.13), we have the following: for all $t$, if $\theta_t, \theta_t'$ are the solutions to eq. (2) using $D, D'$ respectively and a fixed realization of the Brownian motion, then

$$\left\| \theta_t - \theta_t' \right\|_2 \leq \frac{2L}{mn}.$$

Taking the expectation over the Brownian motion and using $L$-Lipschitzness of $\ell$, we can conclude that for any $t$ and all $d \in \tau$,

$$\mathbb{E}_{\theta_t, \theta_t'}[\ell(\theta_t; d) - \ell(\theta_t'; d)] \leq \frac{2L^2}{mn}.$$

$\square$

*Proof of Corollary E.3.* Similarly to the proof of Corollary E.2, if $\theta_t, \theta_t'$ are the solutions to eq. (2) (run on the regularized loss function $\mathcal{L}_m$) using $D, D'$ respectively with $\beta_t = \beta$ and fixed realization of the Brownian motion, then

$$\mathbb{E}_{\theta_t, \theta_t'}[\left\| \theta_t - \theta_t' \right\|_2] \leq \frac{2L}{mn},$$

and thus

$$\mathbb{E}_{\theta_t, \theta_t'}[\ell(\theta_t; d) - \ell(\theta_t'; d)] \leq \frac{2L^2}{mn}.$$

Note that the above inequality holds for the *unregularized* loss. Taking the limit as $t \to \infty$ and use Lemma D.2 to conclude that, for $\theta^{\mathsf{priv}}$ and $\theta^{\mathsf{priv}'}$ output by Algorithm 2 using $D$ and $D'$, respectively,

$$\mathbb{E}_{\theta^{\mathsf{priv}}, \theta^{\mathsf{priv}'}}[\ell(\theta^{\mathsf{priv}}; d) - \ell(\theta^{\mathsf{priv}'}; d)] \leq \frac{2L^2}{mn}$$

for all $d \in \tau$. $\square$

Then, from Corollary E.3 and Theorem D.1, we have the following theorem:

**Theorem E.4.** *Let $\theta^{\text{priv}}$ be the output of Algorithm 2 when run on the regularized loss $\mathcal{L}_m(\theta; D) := \mathcal{L}(\theta; D) + \frac{m}{2} \|\theta\|_2^2$. If each $\ell$ is convex and L-Lipschitz, then for $m = \frac{L}{\|\mathcal{C}\|_2 \sqrt{n}}$ we have:*

$$\mathbb{E}_{\theta^{\text{priv}}} [\text{Risk}_{\text{SCO}}(\theta^{\text{priv}})] = O\left( \frac{Lp \|\mathcal{C}\|_2}{\varepsilon n} + \frac{L \|\mathcal{C}\|_2}{\sqrt{n}} \right).$$

*Furthermore, outputting $\theta^{\text{priv}}$ is $(\varepsilon, \delta)$-differentially private.*

*Proof.* Assume wlog that $\mathbf{0} \in \mathcal{C}$. Let $\theta^*$ be the empirical minimizer of $\mathcal{L}$. If $m \leq \frac{L}{\|\mathcal{C}\|_2}$, the Lipschitz constant does not change by more than a constant factor, so we have by Theorem D.1,:

$$\mathbb{E}_{\theta^{\text{priv}}} [\mathcal{L}_m(\theta^{\text{priv}}; D)] - \mathcal{L}_m(\theta^*; D) \leq \mathbb{E}_{\theta^{\text{priv}}} [\mathcal{L}_m(\theta^{\text{priv}}; D)] - \min_{\theta \in \mathcal{C}} \mathcal{L}_m(\theta; D) = O\left( \frac{Lp \|\mathcal{C}\|_2}{\varepsilon n} \right).$$

The functions $\mathcal{L}_m$ and $\mathcal{L}$ differ by at most $\frac{m}{2} \|\mathcal{C}\|_2^2$ everywhere in $\mathcal{C}$, so in turn:

$$\mathbb{E}_{\theta^{\text{priv}}} [\text{Risk}_{\text{ERM}}(\theta^{\text{priv}})] = O\left( \frac{Lp \|\mathcal{C}\|_2}{\varepsilon n} + m \|\mathcal{C}\|_2^2 \right).$$

Finally, we apply the uniform stability bound from Corollary E.3 to get:

$$\mathbb{E}_{\theta^{\text{priv}}} [\text{Risk}_{\text{SCO}}(\theta^{\text{priv}})] = O\left( \frac{Lp \|\mathcal{C}\|_2}{\varepsilon n} + m \|\mathcal{C}\|_2^2 + \frac{L^2}{mn} \right).$$

The theorem follows by setting $m = \frac{L}{\|\mathcal{C}\|_2 \sqrt{n}}$. $\qquad \square$

### E.2 BOUND FOR STRONGLY CONVEX LOSSES

Under strong convexity, using a similar proof to the proof of Theorem E.4, we can obtain near-optimal DP-SCO bounds for $\varepsilon$-DP.

We first show that the empirical minimizer is close to the population minimizer with high probability:

**Lemma E.5.** *Let $\ell$ be a m-strongly convex function and $\mathcal{C} \subset \mathbb{R}^p$ be a convex set with bounded diameter such that for any $d, \theta$,*

$$\|\nabla \ell(\theta; d) - \mathbb{E}_{d \sim \mathcal{D}} [\nabla \ell(\theta; d)]\|_2 \leq \Delta,$$

*and let $\theta^* := \arg\min_{\theta \in \mathcal{C}} \mathbb{E}_{d \sim \mathcal{D}} [\ell(\theta; d)]$ and $\theta^{emp} := \arg\min_{\theta \in \mathcal{C}} \ell(\theta; D)$. Then for $D \sim \mathcal{D}^n$, with probability $1 - \beta$, we have:*

$$\|\theta^{emp} - \theta^*\|_2 = O\left( \frac{\Delta \sqrt{\log(1/\beta)}}{m \sqrt{n}} \right)$$

*Proof.* Consider a function $\widetilde{\ell}$ over $\mathbb{R}^p$ which has gradient $\nabla\widetilde{\ell}(\theta) = \nabla\ell(\Pi_{\mathcal{C}}(\theta)) + m(\theta - \Pi_{\mathcal{C}}(\theta))$. We will show that the assumptions on $\ell$ in the lemma statement also hold for $\widetilde{\ell}$, and that the distance between the empirical and population minimizers of $\widetilde{\ell}$ over $\mathbb{R}^p$ dominates the distance between the empirical and population minimizers of $\ell$ over $\mathcal{C}$. So proving the lemma holds for $\widetilde{\ell}$ and $\mathbb{R}^p$ implies it holds for $\ell$ and $\mathcal{C}$, i.e. it suffices to prove the lemma assuming $\mathcal{C} = \mathbb{R}^p$.

By $m$-strong convexity of $\ell$ in $\mathcal{C}$, $\widetilde{\ell}$ is also $m$-strongly convex[8]. Also, for any $d, \theta$ we have

$$\|\nabla\ell(\theta; d) - \mathbb{E}_{d\sim\mathcal{D}}[\nabla\ell(\theta; d)]\|_2 = \left\|\nabla\widetilde{\ell}(\theta; d) - \mathbb{E}_{d\sim\mathcal{D}}\left[\nabla\widetilde{\ell}(\theta; d)\right]\right\|_2 \leq \Delta.$$

So, we have shown the assumptions in the lemma hold on $\widetilde{\ell}$ as desired.

From strong convexity of $\widetilde{\ell}$, the empirical/population minimizers of $\widetilde{\ell}$ over $\mathbb{R}^p$ are the unique points with gradient $\mathbf{0}$. This gives us that for any $D$, the empirical minimizer of $\widetilde{\ell}$ over $\mathbb{R}^p$ is equal to

$$\widetilde{\theta}^{emp} := \theta^{emp} - \frac{1}{m} \cdot \nabla\ell(\theta^{emp}; D),$$

where $\theta^{emp}$ is the minimizer of $\ell(\theta; D)$, and the population minimizer of $\mathbb{E}_{d\sim\mathcal{D}}\left[\widetilde{\ell}(\theta; d)\right]$ is

$$\widetilde{\theta}^* := \theta^* - \frac{1}{m} \cdot \mathbb{E}_{d\sim\mathcal{D}}[\nabla\ell(\theta^*; d)].$$

Note that by convexity either $\nabla\ell(\theta^{emp}; D) = \mathbf{0}$ or $-\nabla\ell(\theta^{emp}; D)$ is a tangent vector of $\mathcal{C}$ at $\theta^{emp}$ (and the same is true for $\theta^*$). In turn, $\theta^{emp} = \Pi_{\mathcal{C}}(\widetilde{\theta}^{emp})$ and $\theta^* = \Pi_{\mathcal{C}}(\widetilde{\theta}^*)$, and since projection is a non-expansive operator, $\left\|\widetilde{\theta}^{emp} - \widetilde{\theta}^*\right\|_2 \geq \|\theta^{emp} - \theta^*\|_2$. This shows the distance between minimizers of $\widetilde{\ell}$ dominates the distance between minimizers of $\ell$ as desired.

We now turn to proving the lemma assuming $\mathcal{C} = \mathbb{R}^p$. If $\mathcal{C} = \mathbb{R}^p$ then $\mathbb{E}_{d\sim\mathcal{D}}[\nabla\ell(\theta; d)] = \mathbf{0}$. Now, by the assumptions in the lemma and a vector Azuma inequality (see e.g., Hayes (2003)), we have $\|\nabla\ell(\theta^*; D)\|_2 = O(\frac{\Delta\sqrt{\log(1/\beta)}}{\sqrt{n}})$ with probability $1 - \beta$ over $D$. Furthermore, we know $\nabla\ell(\theta^{emp}; D) = \mathbf{0}$ by strong convexity and since $\mathcal{C} = \mathbb{R}^p$. Then by strong convexity, we have

$$\|\theta^* - \theta^{emp}\|_2 \leq \frac{\|\nabla\ell(\theta^*; D) - \nabla\ell(\theta^{emp}; D)\|_2}{m} = \frac{\|\nabla\ell(\theta^*; D)\|_2}{m} = O(\frac{\Delta\sqrt{\log(1/\beta)}}{m\sqrt{n}})$$

with probability $1 - \beta$ as desired. $\qquad\square$

Given Lemma E.5, if we want to ensure the *population* minimizer rather than empirical minimizer remains in the sets we choose in Algorithm 3, we just need to choose a slightly larger ball. From this modification and uniform stability, we get the following DP-SCO bound:

**Theorem E.6.** *Let $m, n, p, \ell$ be as in lemma E.5. Let $\theta_k$ be the output of Algorithm 3 except we let the radius of $\mathcal{C}_i$ be $\sqrt{\frac{cL(p+3\log n)\|\mathcal{C}_{i-1}\|_2}{m\varepsilon_i n}} + \frac{cL\sqrt{\log n}}{m\sqrt{n}}$ instead of $\sqrt{\frac{cL(p+3\log n)\|\mathcal{C}_{i-1}\|_2}{m\varepsilon_i n}}$ (chosen in Theorem D.3 to obtain optimal ERM bound). Then*

$$\mathbb{E}[\mathsf{Risk}_{\mathsf{SCO}}(\theta_k)] = O\left(\frac{L^2 p^2 \log n}{m\varepsilon^2 n^2} + \frac{L^2}{mn}\right).$$

---

[8]This follows since $\widetilde{\ell}(\theta)$ is $m$-strongly convex iff $\widetilde{\ell}(\theta) - \frac{m}{2}\|\theta\|_2^2$ is. By $m$-strong convexity of $\ell$ in $\mathcal{C}$, $\widetilde{\ell}(\theta) - \frac{m}{2}\|\theta\|_2^2$ is convex in $\mathcal{C}$, and then its convexity over $\mathbb{R}^p$ then follows from e.g. Theorem 4.1 in Yan (2012).

*Proof.* Note that by *L*-Lipschitzness of $\ell$ in $\mathcal{C}$, we have $\|\nabla\ell(\theta;d) - \mathbb{E}_{d\sim\mathcal{D}}[\nabla\ell(\theta;d)]\|_2 \le 2L$. By Lemma D.4, Lemma E.5, and a triangle inequality, we have that if $c$ is sufficiently large then the population minimizer of $\ell$ in $\mathcal{C}_i$ is in $\mathcal{C}_{i+1}$ for each $i$ with probability $1 - 2/n^3$. Then by a union bound, we have that the population minimizer is in $\mathcal{C}_{k-1}$. When this event fails to hold, our excess population loss is $O(L\|\mathcal{C}_0\|_2)$ and so the contribution of this event to the expected excess loss is $O\left(\frac{L\|\mathcal{C}_0\|_2 \log\log(\varepsilon n)}{n^3}\right) = O\left(\frac{L^2 \log\log(\varepsilon n)}{mn^3}\right)$, which is asymptotically less than our desired bound. So it suffices to provide the desired expected excess loss bound conditioned on this event. We can bound the diameter of $\mathcal{C}_{k-1}$ similarly to the proof of Theorem D.3, by noting that:

$$\|\mathcal{C}_i\|_2 \le 2 \cdot \max\left\{ 2\sqrt{\frac{cL(p + 3\log n)\|\mathcal{C}_{i-1}\|_2}{m\varepsilon_i n}}, \frac{cL\sqrt{\log n}}{m\sqrt{n}} \right\}$$

Then, rolling out the recursion, we have similarly to the proof of Theorem D.3:

$$\|\mathcal{C}_{k-1}\|_2 = O\left( \frac{L(p + \log n)}{m\varepsilon n} + \frac{L\sqrt{\log n}}{m\sqrt{n}} \right).$$

Now, combining Theorem D.1 and the uniform stability bound of Corollary E.3, we get that the expected excess population loss of $\theta_k$ compared to the population minimizer over $\mathcal{C}_{k-1}$ is:

$$O\left( \frac{Lp\|\mathcal{C}_{k-1}\|_2}{\varepsilon n} + \frac{L^2}{mn} \right) = O\left( \frac{L^2}{mn} \cdot \left( \frac{(p^2 + p\log n)}{\varepsilon^2 n} + \frac{p\sqrt{\log n}}{\varepsilon\sqrt{n}} + 1 \right) \right) = O\left( \frac{L^2 p^2 \log n}{m\varepsilon^2 n^2} + \frac{L^2}{mn} \right).$$

In the final equality, we use the fact that $\frac{p\sqrt{\log n}}{\varepsilon\sqrt{n}}$ is the geometric mean of $\frac{p^2\log n}{\varepsilon^2 n}$ and 1 and thus $\frac{p\sqrt{\log n}}{\varepsilon\sqrt{n}} \le \max\{\frac{p^2\log n}{\varepsilon^2 n}, 1\}$. We conclude by noting that conditioned on the event the population minimizer over $\mathcal{C}_0$ is contained in $\mathcal{C}_{k-1}$, $\theta_k$ has this same excess population loss bound compared to the population minimizer over $\mathcal{C}_0$. □

# F  LD AS NOISY GRADIENT FLOW AND DP-ERM AND DP-SCO UNDER $(\varepsilon, \delta)$-DP

We now investigate LD (defined in (1) and (2)) as a noisy gradient flow algorithm (i.e., for finite time $T$ or when it is far from convergence). In fact, in Section H, we argue that the setting of parameters we operate with, the algorithm could not have converged to a limiting distribution in any reasonable sense. In the following, we provide optimal DP-ERM and DP-SCO bounds via LD for both convex and strongly convex losses. All the proofs are deferred to Appendix J.

## F.1  DP-ERM BOUNDS FOR CONVEX AND STRONGLY CONVEX LOSSES

We first provide the excess ERM bounds for the LD in (2) for convex losses.

**Theorem F.1.** *Let the inverse temperature $\beta_t = \beta$ for all $t > 0$. Then for an appropriate choice of $\beta, T$, the solution to (2), $\theta^{\mathsf{priv}} = \theta_T$ is $(\varepsilon, \delta)$-differentially private and*

$$\mathbb{E}\left[\mathsf{Risk}_{\mathsf{ERM}}(\theta^{\mathsf{priv}})\right] = O\left(\frac{L\,\|\mathcal{C}\|_2\,\sqrt{p\log(1/\delta)}}{\varepsilon n}\right).$$

The proof of this theorem can be viewed as a continuous analogue of Shamir & Zhang (2013). We sketch the proof here. We define a potential $\phi_t \propto \|\theta_t - \theta^*\|_2^2$, and use Ito's lemma to analyze the rate of the change of $\phi_t$. Rearranging the resulting inequality and using convexity, we then bound the integral of the excess loss of $\theta_t$ over any interval of time $[a, b]$. Using techniques similar to those in Shamir & Zhang (2013), we translate this into a bound on the excess loss of $\theta_T$ in terms of $T$ and $\beta$. We use Lemma C.1 to choose the value of $\beta$ that preserves privacy, and then optimize the resulting bound over $T$.

We now provide the excess ERM bounds of LD in (2) for strongly convex losses. To simplify the presentation, we assume only in this theorem alone $\mathcal{L}$ satisfies $2m$-strong convexity instead of $m$-strong convexity - this does not affect our final bound by more than constant factors.

**Theorem F.2.** *Let $\theta_t$ be the solution to (2) if we set $\beta_t = t^a$. Suppose $\mathcal{L}$ is $2m$-strongly convex. Then for any $\varepsilon, \delta$ and an appropriate choice of $a, T$, for $B_t = \frac{t^{a+1}}{a+1}$, and $\theta^{\mathsf{priv}} = \frac{1}{e^{mB_T}-1}\int_0^T \theta_t de^{mB_t}$, over the randomness of the algorithm,*

$$\mathbb{E}\left[\mathsf{Risk}_{\mathsf{ERM}}(\theta^{\mathsf{priv}})\right] = O\left(\frac{pL^2\log(1/\delta)}{m\varepsilon^2 n^2}\log\left(\frac{\varepsilon n}{L\sqrt{\log(1/\delta)}}\right)\log\left(\frac{p\varepsilon^2 n^2}{\log(1/\delta)}\right)\right).$$

*Furthermore, outputting $\theta^{\mathsf{priv}}$ is $(\varepsilon, \delta)$-differentially private.*

The proof is fairly similar to that of Theorem F.1; the main differences are that we now define the potential $\phi_t$ to be proportional to $e^{mB_t}\|\theta_t - \theta^*\|_2^2$ instead of $\|\theta_t - \theta^*\|_2^2$, and there is no need to translate to a bound on the final iterate like in Theorem F.1.

### F.2 DP-SCO BOUNDS FOR CONVEX AND STRONGLY CONVEX LOSSES

The uniform stability bound of LD in the case of $(\varepsilon, \delta)$-DP follows from taking the limit as the step sizes go to 0 in the bound presented in (Bassily et al., 2020, Lemma 3.1):

**Lemma F.3.** *Let $\theta_T$ be the solution to (2) at time $T$, and suppose each $\ell$ is $L$-Lipschitz. Then outputting $\theta_T$ satisfies $\mu$-uniform stability for: $\mu = \frac{4L^2}{n}\int_0^T \beta_t dt$.*

Combining Lemma F.3 with Theorem F.1, we obtain tight DP-SCO guarantees for convex losses.

**Theorem F.4.** *If each $\ell$ is convex and $L$-Lipschitz, then for an appropriate choice of $\beta_t, T$, for $\theta^{\mathsf{priv}} = \theta_T$ that is the solution to (2), we have:*

$$\mathbb{E}_{\theta^{\mathsf{priv}}}[\mathsf{Risk}_{\mathsf{SCO}}(\theta^{\mathsf{priv}})] = O\left(\frac{L\,\|\mathcal{C}\|_2\,\sqrt{p\log(1/\delta)}}{\varepsilon n} + \frac{L\,\|\mathcal{C}\|_2}{\sqrt{n}}\right).$$

We can also combine Corollary E.2 and Theorem F.2 to obtain tight DP-SCO guarantees for strongly convex losses.

**Theorem F.5.** *Suppose $\ell$ is m-strongly convex and L-Lipschitz, then for an appropriate choice of $\beta_t, T$, for $\theta^{\text{priv}}$ as defined in Theorem F.2, we have:*

$$\mathbb{E}_{\theta^{\text{priv}}}\left[\text{Risk}_{\text{SCO}}(\theta^{\text{priv}})\right] = O\left(\frac{pL^2\log(1/\delta)}{m\varepsilon^2 n^2}\log\left(\frac{\varepsilon n}{L\sqrt{\log(1/\delta)}}\right)\log\left(\frac{p\varepsilon^2 n^2}{\log(1/\delta)}\right) + \frac{L^2}{mn}\right).$$

*In both cases, outputting $\theta^{\text{priv}}$ is $(\varepsilon, \delta)$-differentially private.*

## G LOWER BOUND ON DP-ERM FOR NON-CONVEX LOSSES

In this section, we show the following lower bound on the excess empirical risk for 1-Lipschitz non-convex loss functions. The lower bound implies that that there is no advantage, in terms of the dependence on dimensions ($p$), to move from $\varepsilon$-DP to $(\varepsilon, \delta)$-DP.

**Theorem G.1.** *Let $\varepsilon \leq 1$, $2^{-\Omega(n)} \leq \delta \leq 1/n^{1+\Omega(1)}$, and $B(\mathbf{0}, 1)$ be a unit Euclidean ball centered at origin. Then there exists 1-Lipschitz non-convex function $\mathcal{L} : B(0,1) \times \mathcal{X} \to \mathbb{R}$ and a dataset [9] $D = \{d_1, \cdots, d_n\}$ such that for every $p \in \mathbb{N}$, there is no $(\varepsilon, \delta)$-differentially private algorithm $\mathcal{A}$ that outputs $\theta^{\text{priv}}$ such that*

$$\mathbb{E}\left[\mathcal{L}(\theta^{\text{priv}}; D) - \min_{\theta \in \mathbb{B}_p(1)} \mathcal{L}(\theta; D)\right] = o\left(\frac{p\log(1/\delta)}{n\varepsilon}\right), \tag{12}$$

*Proof.* We first perform two translations of Theorem B.12: first from $(1, \frac{1}{ns})$ to $(\varepsilon, \delta)$ from Steinke & Ullman (2015) and then from sample complexity to a result stated in the terms of accuracy bound. A direct corollary of Theorem B.12 with $k = 1$ is as follows: for every $s \in \mathbb{N}$, no $(\varepsilon, \delta)$-differentially private algorithm on input $X$ satisfying the premise of Theorem B.12 outputs an index $j \in [s]$ such that

$$\mathbb{E}_{\mathcal{M}}\left[\frac{1}{n}\sum_{i=1}^{n}\mathbf{x}_{i,j}\right] - \max_{u \in [s]}\frac{1}{n}\sum_{i=1}^{n}\mathbf{x}_{i,u} = o\left(\frac{1}{n\varepsilon}\log(s)\log(1/\delta)\right), \tag{13}$$

where $\varepsilon \leq 1$ and $2^{-\Omega(n)} \leq \delta \leq 1/n^{1+\Omega(1)}$.

Using this lower bound on top-selection, we give our lower bound by defining an appropriate non-convex loss function. In particular, we define a packing over the $p$-dimensional Euclidean ball such that there is an bijective mapping between the centers of the packing and $[s]$. Then the function attains the minimum at the center of packing which corresponds to the coordinate $j \in [s]$ with maximum frequency. Since the size of the $\alpha$-net is $\approx 1/\alpha^p$ and there is a bijective mapping, this gives a lower bound using eq. (13).

Let $B(\mathbf{0}, 1)$ be the $p$-dimensional Euclidean ball centered at origin and let $\alpha \in (0, 1/2)$ be a constant. Consider an $\alpha$-packing with centers $C = \{\mathbf{c}_1, \mathbf{c}_2, \cdots, \}$. It is known that the size of such packing,

---

[9]The dataset, $D = \{d_1, \cdots, d_n\}$ is such that $d_i \in \{0, 1\}^s$ for all $i \in [n]$, $d_{i,j}$ is independent (conditioned on $P$) and $\mathbf{E}[d_{i,j}] = P^j$ for all $i \in [n]$ and $j \in [s]$. Here $\mathcal{P}$ is the distribution that is defined in Theorem B.12.

$N(\alpha)$ is $\left(\frac{1}{\alpha}\right)^p \leq N(\alpha) \leq \left(\frac{3}{\alpha}\right)^p$. Let $s = N(\alpha)$. Further, let $f : B(\mathbf{0}, 1) \to \{1, \cdots, s\}$ be an injective function defined as follows:

$$f(\theta) = \left\{ j : \mathbf{c}_j = \arg\min_{\mathbf{c} \in C} \|\theta - \mathbf{c}\|_2 \right\}.$$

In particular, $f$ is the function that maps a point on the unit ball to its closest point in $C$.

We now define our loss function as follows:

$$\mathcal{L}(\theta; D) := \frac{1}{n} \sum_{d_i \in D} \ell(\theta; d_i) \text{ where } \ell(\theta; d_i) = \min_{\mathbf{c}_j \in C} \left( \frac{\|\theta - \mathbf{c}_j\|}{\alpha} - 1 \right) d_{i,j}. \tag{14}$$

For Lipschitz property, note that each loss function is $1/\alpha$-Lipschitz because the gradient when it is defined is just $\frac{\theta - \mathbf{c}_j}{\alpha \|\theta - \mathbf{c}_j\|_2}$. We prove it formally.

Consider any $\theta, \theta'$ in $B(\mathbf{0}, 1)$ and a data point $d_i \in D$. We wish to show $|\ell(\theta; d_i) - \ell(\theta'; d_i)| \leq \frac{1}{\alpha} \|\theta - \theta'\|_2$. We can split the line segment from $\theta$ to $\theta'$ into a sequence of line segments $(\theta_0, \theta_1), (\theta_1, \theta_2), \ldots, (\theta_{k-1}, \theta_k)$ where $\theta_0 = \theta, \theta_k = \theta'$, such that for any line segment $(\theta_m, \theta_{m+1})$, $\theta_m$ and $\theta_{m+1}$ share a minimizer in $C$ of $\left( \frac{\|\theta - \mathbf{c}_j\|_2}{\alpha} \right) d_{i,j}$.[10]

It now suffices to show $|\ell(\theta_m; d_i) - \ell(\theta_{m+1}; d_i)| \leq \frac{1}{\alpha} \|\theta_m - \theta_{m+1}\|_2$ for each $m$, since we then have:

$$|\ell(\theta; d_i) - \ell(\theta'; d_i)| \leq \sum_{m=0}^{k-1} |\ell(\theta_m; d_i) - \ell(\theta_{m+1}; d_i)| \leq \frac{1}{\alpha} \sum_{m=0}^{k-1} \|\theta_m - \theta_{m+1}\|_2 = \frac{1}{\alpha} \|\theta - \theta'\|_2.$$

Let $\mathbf{c}_j$ be a shared minimizer of $\left( \frac{\|\theta - \mathbf{c}_j\|_2}{\alpha} \right) d_{i,j}$ for $\theta_m$ and $\theta_{m+1}$. If $d_{i,j} = 0$, then trivially $|\ell(\theta_m; d_i) - \ell(\theta_{m+1}; d_i)| \leq \frac{1}{\alpha} \|\theta_m - \theta_{m+1}\|_2$. Otherwise $d_{i,j} = 1$ and by triangle inequality, we have:

$$|\ell(\theta_m; d_i) - \ell(\theta_{m+1}; d_i)| = \left| \frac{\|\theta_m - \mathbf{c}_j\|_2}{\alpha} - \frac{\|\theta_{m+1} - \mathbf{c}_j\|_2}{\alpha} \right| \leq \frac{1}{\alpha} \|\theta_m - \theta_{m+1}\|_2.$$

Now let us suppose there is an $(\varepsilon, \delta)$-differentially private algorithm $\mathcal{A}$ that on input a non-convex function $\mathcal{L}$ and $n$ data points $\{d_1, \cdots, d_n\}$, outputs a $\theta^{\text{priv}}$ such that

$$\mathbb{E}_{\mathcal{A}} \left[ \mathcal{L}(\theta^{\text{priv}}; D) \right] - \min_{\theta \in B(1)} \mathcal{L}(\theta; D) = o \left( \frac{p \log (1/\delta)}{n\varepsilon} \right), \tag{15}$$

where $D = \{d_1, \cdots, d_n\}$.

---

[10]In particular, for each $\mathbf{c}_j$ let $B_j$ be the set of points in $B(\mathbf{0}, 1)$ such that $\mathbf{c}_j$ is a minimizer of $\left( \frac{\|\theta - \mathbf{c}_j\|_2}{\alpha} \right) d_{i,j}$.
We can split the line segment from $\theta$ to $\theta'$ at each point where it enters or leaves some $B_j$ to get this sequence of line segments, and by this construction each line segment's endpoints are both in $B_j$ for some $j$.

We will construct an algorithm that uses $\mathcal{A}$ as subroutine and solve top-selection problem with an error $o(\log(s))$, contracting the lower bound of Theorem B.12.

Algorithm $\mathcal{B}$:

- On input $X = \{\mathbf{x}_1, \cdots, \mathbf{x}_n\}$, invokes $\mathcal{A}$ on the function defined by eq. (14) and data points $X$ to get $\theta^{\mathsf{priv}}$ as output.
- Output $f(\theta^{\mathsf{priv}})$.

Since the last step is post-processing, $\mathcal{B}$ is $(\varepsilon, \delta)$-differentially private. We now show that if $\mathcal{A}$ outputs a $\theta^{\mathsf{priv}}$ satisfying eq. (15), then $j := f(\theta^{\mathsf{priv}})$ satisfies eq. (13) leading to a contradiction.

First note that, for any $\mathbf{c} \in C$ and all $\theta \in \mathbb{B}_p(\mathbf{c}, \alpha)$ such that $\|\theta - \mathbf{c}\|_2 \leq \frac{\alpha}{2}$,

$$\mathcal{L}(\mathbf{c}; D) = -\frac{1}{n} \sum_{i=1}^{n} \mathbf{x}_{i, f(\mathbf{c})} \leq \mathcal{L}(\theta; D).$$

Therefore,

$$\mathcal{L}(\theta^*; X) := \min_{\mathbf{c} \in C} \mathcal{L}(\mathbf{c}, X) = \min_{\mathbf{c} \in C} \left( -\frac{1}{n} \sum_{i=1}^{n} \mathbf{x}_{i, f(\mathbf{c})} \right)$$

This implies that

$$f(\theta^*) = \arg\max_{1 \leq j \leq s} \frac{1}{n} \sum_{i=1}^{n} \mathbf{x}_{i,j},$$

which is exactly the top-selection problem. Therefore, eq. (15) implies eq. (13) because $p \log \left( \frac{1}{\alpha} \right) \leq \log(s) \leq p \log \left( \frac{3}{\alpha} \right)$ and $\alpha \in (0, 1/2)$ is a constant.

$\square$

## H  ON THE CONVERGENCE TIME OF LD UNDER $(\varepsilon, \delta)$-DP

In this section we provide a discussion into the choice of the time for which LD was run for our approximate DP results. In particular, we will discuss why for convex losses, the optimal runtime in Theorem F.1 is $\propto 1/p$, and why for the setting of parameters in Theorem F.1, the diffusion process does not come close to converging to the stationary distribution. We show that such properties are present even in DP-SGD (Bassily et al., 2014), when analyzed carefully while taking the learning rate and the number of time steps into account.

**On Optimal Time for DP-ERM with Approximate DP:** For DP-ERM with a convex loss and approximate DP, our eventual choice of $T$ is $\|\mathcal{C}\|_2^2 / p$, i.e. decreasing in the dimension if $\|\mathcal{C}\|_2$ is fixed. We note that this phenomenon is not unique to our analysis; e.g., for learning rate $\eta > 0$, the eventual value of $T\eta$ in the same setting in Bassily et al. (2019) also is roughly proportional to $1/p$. This is perhaps counterintuitive, as in the non-noisy setting, the amount of time one runs gradient descent for is generally independent of the dimension. We can provide some intuition for this phenomenon from the perspective of the Langevin diffusion.

As an example, consider the loss function $\mathcal{L}(\theta) = \|\theta\|_2$, and suppose $\mathcal{C} = B(\mathbf{0}, 1)$, i.e. an $\ell_2$ ball of radius 1 centered at the origin. For privacy, per Lemma C.1 we will set $\beta \propto 1/\sqrt{T} = \sqrt{p}$. At

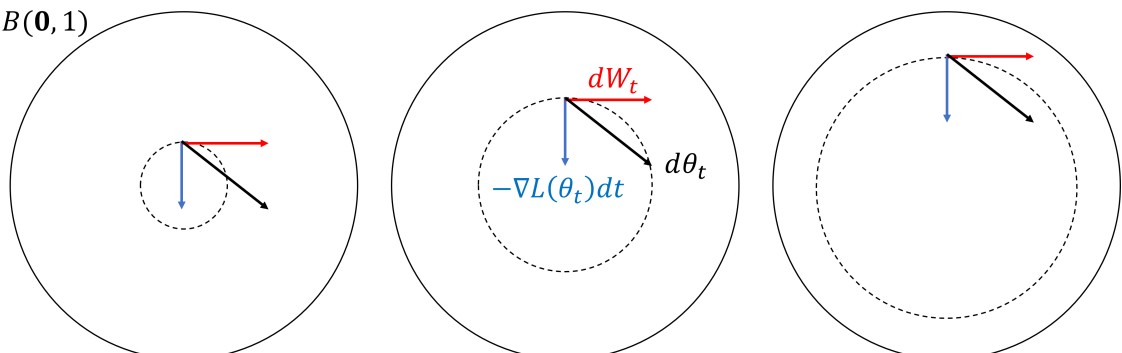

Figure 1: An abstract visualization of how the gradient drift and Brownian motion counteract each other in Langevin diffusion for $\mathcal{L}(\theta; D) = \|\theta\|_2$, $\mathcal{C} = B(\mathbf{0}, 1)$ is the unit Euclidean ball centered at $\mathbf{0}$. The blue arrow shows the integral of gradient drift over time $\eta$. The red arrow shows the integral of Brownian motion over time $\eta$ perpendicular to the gradient drift. In the left picture, we can see that when we are close to the minimizer (the center of the circle), Brownian motion has a stronger effect than gradient drift. In the right picture, we see that when we are far from the minimizer, the opposite is true. In the middle picture, we see the equilibrium point where the two counteract each other. As the dimension increases, the ratio of the red arrow's length to blue arrow's length increases, and so the equilibrium point will get further from the minimizer.

distance proportional to $p/\beta \propto \sqrt{p}$ from the minimizer, Langevin diffusion stops making progress towards the minimizer in expectation, as at this distance the progress towards the minimizer due to the drift $-\nabla \mathcal{L}(\theta)$ is cancelled out by the movement in perpendicular directions due to the Brownian motion (see Figure 1 for a visualization of this phenomenon). This suggests that we only need to run the diffusion until $\theta_t$ reaches this distance from the minimizer, as past that point, we do not expect the diffusion to make progress. Now, the distance $\sqrt{p}$ is increasing with the dimension, and in turn the distance from the ball of radius $\sqrt{p}$ centered at the origin to the boundary of $\mathcal{C}$ is *decreasing* since $\|\mathcal{C}\|_2$ is a constant. So even if we start at the boundary of $\mathcal{C}$ (the worst case for this loss function) the distance the Langevin diffusion needs to travel to be within $\sqrt{p}$ distance of the origin is decreasing with the dimension. The total distance we travel due to the gradient drift is roughly $T\beta \propto \sqrt{T}$, and thus the time we need to run the diffusion for to reach this ball also decreases with the dimension. In particular, once the dimension is large enough, the diffusion will stay very close to the boundary of $\mathcal{C}$ with high probability, i.e. taking an arbitrary initial point and outputting it is about as good as in this example as running the diffusion for any amount of time.

**On Non-Convergence of DP-SGD/Finite Time Diffusion:** We show here that while both the Langevin diffusion and DP-SGD achieve optimal error rates in finite time, for the same parameter settings they do not converge to the stationary distribution. This in part explains why e.g. the finite-time Langevin diffusion for convex losses achieves a much better privacy parameter than its stationary distribution. We use the same example as in the preceding discussion, i.e. $\mathcal{L}(\theta) = \|\theta\|_2$, $\mathcal{C} = B(\mathbf{0}, 1)$. In the proof of Theorem F.1, we show optimal error rates are achieved when Langevin diffusion runs for time $T = \frac{1}{2p}$. Asymptotically, we get the same bound if we use e.g. $T = \frac{1}{100p}$, so let us instead consider this choice of $T$. Since $\beta \propto 1/\sqrt{T}$, and the movement due to the gradient drift is proportional to $T\beta$, the movement due to gradient drift for our parameter choices decreases with $p$. The integral from 0 to $T$ of the Brownian motion has distribution $N(\mathbf{0}, T\mathbb{I}_p)$, i.e.

with probability $1 - e^{-\Omega(p)}$, the total movement due to Brownian motion is at most, say, $2\sqrt{Tp} = \frac{1}{5}$. In particular, we get that for all sufficiently large $p$, with probability $1 - e^{-\Omega(p)}$, Langevin diffusion does not move more than a total distance $\frac{1}{3}$. If $\theta_0$ is on the boundary, then the probability a random point from the stationary distribution is in a ball of radius $\frac{1}{3}$ centered at $\theta_0$ is $e^{-\Omega(p)}$. So even though $\theta_T$ in expectation obtains the (asymptotic) optimal excess empirical loss, the total variation distance between $\theta_T$ and the stationary distribution is at least $1 - e^{-\Omega(p)}$, and similarly e.g. any $p$-Wasserstein distance between $\theta_T$ and the stationary distribution is $\Omega(1) = \Omega(\|\mathcal{C}\|_2)$.

# I    SPLIT REGIMES FOR RÉNYI DIVERGENCE BOUNDS ON LANGEVIN DIFFUSION

In this section, we show that when the loss functions are strongly convex and smooth, we can show a bound on the Rényi divergence between two diffusions using different loss functions that converges to roughly the divergence between the stationary distributions of the diffusions. In doing so, we show that Rényi divergence bounds between two diffusions exist in two different regimes, where for small $T$ it is advantageous to analyze privacy of LD as a noisy optimization algorithm, and for large $T$ it is advantageous to analyze privacy of LD to an algorithm which samples from approximately a Gibbs distribution.

## I.1    "LONG TERM" RÉNYI DIVERGENCE BOUND

Under $m$-strong convexity and $M$-smoothness, using the results in Vempala & Wibisono (2019) we can also give a bound on the Rényi divergence depending on the closeness to the stationary distribution. Since we rely on the bounds in Vempala & Wibisono (2019) which apply to the *unconstrained* setting, we will also focus only on the unconstrained setting here. In order to ensure the initial Rényi divergence to the stationary distribution is finite, we assume that both $\mathcal{L}(\theta; D), \mathcal{L}(\theta; D')$ have minimizers $\theta_{\mathsf{opt}}, \theta'_{\mathsf{opt}}$ respectively such that $\|\theta_{\mathsf{opt}}\|_2, \|\theta'_{\mathsf{opt}}\|_2 \leq R$.

**Lemma I.1.** *Suppose we sample $\theta_0 = \theta'_0$ from $\Theta_0 = N(\mathbf{0}, \frac{1}{\beta m}\mathbb{I}_p)$. Let $\Theta_t, \Theta'_t$ be the resulting distributions of $\theta_t, \theta'_t$ according to (1) using $D, D'$ respectively. Let $\mathcal{P}, \mathcal{P}'$ be the stationary distributions of (1) using $D, D'$ respectively. Then for any $T \geq t_0 := 2\log((\alpha - 1)\max\{2, M/m\})$ and $\alpha \geq 2$:*

$$R_\alpha(\Theta_T, \Theta'_T) \leq O\left(\beta m R^2((M/m)^2 + \alpha) + \frac{pM}{m}\log\left(\frac{M}{m}\right)\right) \cdot \exp\left(-\frac{(T - t_0)\beta m}{3\alpha}\right) + \frac{4}{3}R_{3\alpha}(\mathcal{P}, \mathcal{P}').$$

*Proof.* Since adding a constant to $\mathcal{L}$ does not affect the sampling problem, assume without loss of generality that there is a density function $\mathcal{P}$ such that $\mathcal{P}(\theta) = \exp(-\beta\mathcal{L}(\theta; D))$. Let $\mathcal{Q} = N(\theta_{\mathsf{opt}}, \frac{1}{\beta m}\mathbb{I}_p)$. By $m$-strong convexity of $\mathcal{L}$, $\mathcal{L}(\theta; D) - \mathcal{L}(\theta_{\mathsf{opt}}; D) \geq \frac{m}{2}\|\theta - \theta_{\mathsf{opt}}\|_2^2$, so we have:

$$\exp((\alpha - 1)R_\alpha(\mathcal{P}, \mathcal{Q})) = \int_{\mathbb{R}^p} \frac{\mathcal{P}(\theta)^\alpha}{\mathcal{Q}(\theta)^{\alpha-1}} d\theta$$

$$= \left(\frac{2\pi}{\beta m}\right)^{p(\alpha-1)/2} \int_{\mathbb{R}^p} \exp(\alpha \log \mathcal{P}(\theta) + \frac{(\alpha - 1)\beta m}{2} \|\theta - \theta_{\mathsf{opt}}\|_2^2) d\theta$$

$$\leq \mathcal{P}(\theta_{\mathsf{opt}})^\alpha \left(\frac{2\pi}{\beta m}\right)^{p(\alpha-1)/2} \int_{\mathbb{R}^p} \exp(-\frac{\beta m}{2} \|\theta - \theta_{\mathsf{opt}}\|_2^2) d\theta$$

$$= \mathcal{P}(\theta_{\mathsf{opt}})^\alpha \left(\frac{2\pi}{\beta m}\right)^{p\alpha/2}$$

$$\leq \left(\frac{\beta M}{2\pi}\right)^{p\alpha/2} \left(\frac{2\pi}{\beta m}\right)^{p\alpha/2} = \left(\frac{M}{m}\right)^{p\alpha/2}.$$

where the first inequality uses strong convexity, and in the second inequality, we use the fact that the $\beta m$-log strongly convex, $\beta M$-log smooth distribution with mode $\theta_{\mathsf{opt}}$ that has the largest density function at $\theta_{\mathsf{opt}}$ is $N(\theta_{\mathsf{opt}}, \frac{1}{\beta M}\mathbb{I}_p)$.

The above bound thus implies that

$$R_\alpha(\mathcal{P}, \mathcal{Q}) \leq \frac{p\alpha}{2(\alpha - 1)} \log\left(\frac{M}{m}\right).$$

By a similar argument, but instead using $M$-smoothness and the fact that the $\beta m$-log strongly convex, $\beta M$-log smooth distribution with mode $\theta_{\mathsf{opt}}$ that has the smallest density function at $\theta_{\mathsf{opt}}$ is $N(\theta_{\mathsf{opt}}, \frac{1}{\beta m}\mathbb{I}_p)$, for $\alpha < \frac{M}{M-m}$:

$$\exp\left((\alpha - 1)R_\alpha(\mathcal{Q}, \mathcal{P})\right) = \int_{\mathbb{R}^p} \frac{\mathcal{Q}(\theta)^\alpha}{\mathcal{P}(\theta)^{\alpha-1}} d\theta$$

$$= (\frac{\beta m}{2\pi})^{\alpha p/2} \int_{\mathbb{R}^p} \exp\left(-\alpha\beta m \frac{\|\theta - \theta_{\mathsf{opt}}\|_2^2}{2} + (\alpha - 1) \log \mathcal{P}(\theta)\right) d\theta$$

$$\leq (\frac{\beta m}{2\pi})^{\alpha p/2} \mathcal{P}(\theta_{\mathsf{opt}})^{-(\alpha-1)} \int_{\mathbb{R}^p} \exp\left((\alpha\beta(M - m) - \beta M)\frac{\|\theta - \theta_{\mathsf{opt}}\|_2^2}{2}\right) d\theta$$

$$\leq (\frac{\beta m}{2\pi})^{p/2} \int_{\mathbb{R}^p} \exp\left((\alpha\beta(M - m) - \beta M)\frac{\|\theta - \theta_{\mathsf{opt}}\|_2^2}{2}\right) d\theta$$

$$\overset{(*)}{=} (\frac{m}{M - \alpha(M - m)})^{p/2}$$

$$\implies R_\alpha(\mathcal{Q}, \mathcal{P}) \leq \frac{p}{2(\alpha - 1)} \log\left(\frac{m}{M - \alpha(M - m)}\right).$$

In $(*)$, we use the assumption on $\alpha$ to ensure the integral does not diverge. In particular, choosing $\alpha = 1 + \frac{m}{M}$ satisfies the assumption and gives:

$$R_{1+\frac{m}{M}}(Q, P) \leq \frac{pM}{2m} \log\left(\frac{M}{m}\right).$$

The same bounds hold for $\mathcal{P}', \mathcal{Q}'$ defined using $D'$ and $\theta'_{\mathsf{opt}}$ instead of $D, \theta_{\mathsf{opt}}$ respectively.

Using the weak triangle inequality for Renyi divergences (Fact B.10), letting $\Theta_0 = N(0, \frac{1}{\beta m}\mathbb{I}_p)$, we have for any $\alpha \geq 1, q \geq 1$:

$$R_\alpha(\mathcal{P}', \Theta_0) \leq \frac{\alpha - 1/q}{\alpha - 1} R_{q\alpha}(\mathcal{P}', \mathcal{Q}') + R_{\frac{q\alpha-1}{q-1}}(\mathcal{Q}', \Theta_0).$$

Setting $q = 2$, plugging in our above bound and Lemma B.8 we get:

$$R_\alpha(\mathcal{P}', \Theta_0) \leq \frac{p\alpha}{2(\alpha - 1)} \log\left(\frac{M}{m}\right) + \frac{\beta m(2\alpha - 1)R^2}{2}.$$

If $\alpha \geq 2$, this can be simplified to:

$$R_\alpha(\mathcal{P}', \Theta_0) \leq p \log\left(\frac{M}{m}\right) + \frac{\beta m(2\alpha - 1)R^2}{2}.$$

Then following the proof in (Ganesh & Talwar, 2020, Lemma 20), we get that if $\alpha \geq 2$:

$$R_\alpha(\mathcal{P}', \Theta'_T) \leq R_\alpha(\mathcal{P}', \Theta_0) \cdot \exp(-\frac{T\beta m}{\alpha}) \leq \left[p \log\left(\frac{M}{m}\right) + \frac{\beta m(2\alpha - 1)R^2}{2}\right] \cdot \exp\left(-\frac{T\beta m}{\alpha}\right)$$

Similarly we have:

$$R_\alpha(\Theta_0, \mathcal{P}) \leq \frac{\alpha - 1/q}{\alpha - 1} R_{q\alpha}(\Theta_0, \mathcal{Q}) + R_{\frac{q\alpha-1}{q-1}}(\mathcal{Q}, \mathcal{P}).$$

Letting $q = \max\{2, M/m\}, \alpha = 1 + \frac{q-1}{q^2}$ and again plugging in our above bound and Lemma B.8:

$$R_{1+\frac{q-1}{q^2}}(\Theta_0, \mathcal{P}) \leq (1+q)R_{q+1-1/q}(\Theta_0, \mathcal{Q}) + R_{1+1/q}(\mathcal{Q}, \mathcal{P})$$

$$\leq (1+q)R_{q+1-1/q}(\Theta_0, \mathcal{Q}) + R_{1+m/M}(\mathcal{Q}, \mathcal{P})$$

$$\leq \frac{\beta m(q^2 + 2q - 1/q)R^2}{2} + \frac{pM}{2m} \log\left(\frac{M}{m}\right).$$

Now, using (Vempala & Wibisono, 2019, Theorem 2 and Lemma 14), we have that for any $\alpha \geq 2$ and $T \geq t_0 := 2\log((\alpha - 1)q)$:

$$R_\alpha(\Theta_T, \mathcal{P}) \leq R_\alpha(\Theta_{t_0}, \mathcal{P}) \cdot \exp\left(-\frac{2t\beta m}{\alpha}\right) \leq \frac{(q-1)}{q^2(1-1/\alpha)} \cdot R_{1+\frac{q-1}{q^2}}(\Theta_0, \mathcal{P}) \cdot \exp\left(-\frac{2(T-t_0)\beta m}{\alpha}\right).$$

To simplify, we will use the fact that $q \geq 2, \alpha \geq 2$ in conjunction with our bound on $R_{1+\frac{q-1}{q^2}}(\Theta_0, \mathcal{P})$ to upper bound this by:

$$\leq \left[\frac{\beta m(q^2 + 2q - 1/q)R^2}{2} + \frac{pM}{2m}\log\left(\frac{M}{m}\right)\right] \cdot \exp\left(-\frac{(T-t_0)\beta m}{\alpha}\right).$$

We can now use the weak triangle inequality (Fact B.10) twice and monotonicity of Renyi divergences (Fact B.6) to directly prove a divergence bound between $\Theta_T$ and $\Theta_T'$ for $\alpha \geq 2$:

$$\begin{aligned}
R_\alpha(\Theta_T, \Theta_T') &\leq \frac{\alpha - 1/3}{\alpha - 1}R_{3\alpha}(\Theta_T, \mathcal{P}) + \frac{3\alpha/2 - 1}{3\alpha/2 - 3/2}R_{3\alpha-1}(\mathcal{P}, \mathcal{P}') + R_{3\alpha-2}(\mathcal{P}', \Theta_T') \\
&\leq \frac{5}{3}R_{3\alpha}(\Theta_T, \mathcal{P}) + \frac{4}{3}R_{3\alpha}(\mathcal{P}, \mathcal{P}') + R_{3\alpha}(\mathcal{P}', \Theta_T')
\end{aligned} \tag{16}$$

Substituting the bounds on $R_{3\alpha}(\Theta_T, \mathcal{P}), R_{3\alpha}(\mathcal{P}, \mathcal{P}')$, and $R_{3\alpha}(\mathcal{P}', \Theta_T')$, we have a bound

$$\begin{aligned}
(16) \leq &\left[\beta m R^2 \cdot \frac{\max\{30, 5(M/m)^2 + 10M/m\} + 18\alpha - 3}{6} + \frac{5pM/m + 3p}{6}\log\left(\frac{M}{m}\right)\right] \\
&\cdot \exp\left(-\frac{(T-t_0)\beta m}{3\alpha}\right) + \frac{4}{3}R_{3\alpha}(\mathcal{P}, \mathcal{P}').
\end{aligned}$$

This completes the proof of Lemma I.1. □

One can modify the proof such that the bound converges to $R_\alpha(\mathcal{P}, \mathcal{P}')$ rather than $\frac{4}{3}R_{3\alpha}(\mathcal{P}, \mathcal{P}')$. To do so, note that both the leading $4/3$ and the leading constant in $3\alpha$ arise from applying Fact B.10 in (16) with fixed parameter $q$. By instead applying Fact B.10 with parameter $q$ depending on $T$, we can replace $\frac{4}{3}R_{3\alpha}(\mathcal{P}, \mathcal{P}')$ with the expression $c(T)R_{\alpha(T)}(\mathcal{P}, \mathcal{P}')$ for some functions $c(T), \alpha(T)$ approaching 1 and $\alpha$ respectively. The cost of doing this is that the coefficient $5/3$ in front of the first term in (16), and the orders $3\alpha$ in the first and last term in (16) will become larger over time; however, for a fixed order $\alpha$ these terms decay as $\exp(-T/\alpha)$, so if we choose $q$ in our application of Fact B.10 such that these values grow sub-linearly, the first and last term in (16) will still decay as $\exp(-T^c)$ for some $0 < c < 1$.

This modification heavily complicates the proof and the form of the final bound, so we omit a proof including this modification and instead opt for a simpler presentation and weaker bound here.

### I.2 SWITCHING BETWEEN "SHORT" AND "LONG" TERM

If both loss functions are $m$-strongly convex and we use a fixed $\beta_t = \beta$, then Chourasia et al. (Chourasia et al., 2021, Corollary 1) implies the following:

**Lemma I.2.** *Suppose* $\|\nabla\mathcal{L}(\theta;D) - \nabla\mathcal{L}(\theta;D')\|_2 \leq \Delta$ *for all* $\theta$, *and* $\mathcal{L}(\theta;D), \mathcal{L}(\theta;D')$ *are both m-strongly convex with respect to* $\theta$. *Then if* $\theta_0$ *and* $\theta'_0$ *are sampled from* $\Theta_0 = N(0, \frac{1}{\beta m}\mathbb{I}_p)$, *for all* $\alpha \geq 1$ *and* $T \geq 0$:

$$R_\alpha(\Theta_T, \Theta'_T) \leq \frac{\alpha\beta\Delta^2}{m} \cdot (1 - e^{-\beta mT/2}).$$

When we have *m*-strongly convexity, *M*-smoothness, and a bound of $\Delta$ on the $\ell_2$-norm of the difference between the gradients of $\mathcal{L}(\theta;D)$ and $\mathcal{L}(\theta;D')$, both Lemma I.1 and Lemma I.2 give a bound on the Renyi divergence. Intuitively, Lemma I.2 bounds the divergence between the noisy gradient flows. Therefore, it is initially 0 when there is no history of the noisy gradient flows and the distributions are the same. However, it worsens as $T$ increases, since the distribution at time $T$ becomes more dependent on the difference between the history of the noisy gradient flows and less dependent on the shared initial distribution. On the other hand, Lemma I.1 effectively is a bound on the "sampling error" between the finite-time distribution and stationary distribution of each diffusion, plus the divergence between the stationary distributions of the two diffusions. The former improves as $T$ increases, and the latter is fixed. In turn, as long as

$$\frac{4}{3}R_{3\alpha}(\mathcal{P}, \mathcal{P}') \leq \frac{\alpha\beta\Delta^2}{m}.$$

That is, the asymptotic bound of Lemma I.2 is worse than the asymptotic bound of Lemma I.1. Since the sampling error goes to 0 as $T$ goes to infinity, there is a shift in regimes where roughly speaking the divergence due to sampling error becomes smaller than the divergence between the noisy gradient flows. At this point, it directs us to use the "long" term bound instead of "short" term bound. In particular, this regime shift occurs roughly when:

$$O(\beta mR^2((M/m)^2 + \alpha) + \frac{pM}{m}\log\left(\frac{M}{m}\right)) \cdot \exp\left(-\frac{(T-t_0)\beta m}{3\alpha}\right) \leq \frac{\alpha\beta\Delta^2}{m} - \frac{4}{3}R_{3\alpha}(\mathcal{P}, \mathcal{P}'),$$

Rearranging, we get that we shift regimes at roughly a time $T^*$ such that

$$T^* \approx 2\log((\alpha-1)\max\{2, M\}) + \Theta\left(\frac{\alpha}{\beta m} \cdot \log\left(\frac{\beta mR(M/m + \alpha) + \frac{pM}{m}\log\left(\frac{M}{m}\right)}{\frac{\alpha\beta\Delta^2}{m} - \frac{4}{3}R_{3\alpha}(\mathcal{P}, \mathcal{P}')}\right)\right).$$

Again, we remark that one can modify Lemma I.1 to approach $R_\alpha(\mathcal{P}, \mathcal{P}')$ instead of $\frac{4}{3}R_\alpha(\mathcal{P}, \mathcal{P}')$. For this asymptotic bound, the inequality

$$R_\alpha(\mathcal{P}, \mathcal{P}') \leq \frac{\alpha\beta\Delta^2}{m},$$

is always satisfied, i.e., either the limiting bound of Lemma I.2 is tight for the stationary distribution, or such a regime shift always exists.

## J  DEFERRED PROOFS FROM SECTION F

*Proof of Theorem F.1.* Let $\phi_{\mathbf{x}}(\theta_t) = \frac{1}{2\beta}\|\theta_t - \mathbf{x}\|_2^2$ be the potential w.r.t. a fixed $\mathbf{x} \in C$. We will write $d\phi_{\mathbf{x}}(\theta_t)$ as the sum of two terms, $d\phi_{\mathbf{x}}^A(\theta_t)$, which is equal to $d\phi_{\mathbf{x}}(\theta_t)$ if we used eq. (1) instead of

eq. (2), and $d\phi_{\mathbf{x}}^B(\theta_t)$, which is simply the difference $d\phi_{\mathbf{x}}(\theta_t) - d\phi_{\mathbf{x}}^A(\theta_t)$. For example, $d\phi_{\mathbf{x}}^B(\theta_t)$ is 0 when $\theta_t$ is not on the boundary of $\mathcal{C}$. Since $\mathcal{C}$ is convex, if $\theta$ is a point on the boundary of $\mathcal{C}$ and $\mathbf{t}$ is a normal vector of $\mathcal{C}$ at $\theta$, then $\langle \theta - \mathbf{x}, \mathbf{t} \rangle \geq 0$. Then by definition of $\Pi_{\mathcal{C},\theta_t}$, this implies $d\phi_{\mathbf{x}}^B(\theta_t) \geq 0$ always, so $d\phi_{\mathbf{x}}(\theta_t) \leq d\phi_{\mathbf{x}}^A(\theta_t)$.

By Ito's lemma (Lemma B.1), we have the following:

$$d\phi_{\mathbf{x}}^A(\theta_t) = \frac{p}{\beta}dt + \left\langle \theta_t - \mathbf{x}, -\nabla\mathcal{L}(\theta; D)dt + \frac{\sqrt{2}}{\beta}dW_t \right\rangle. \tag{17}$$

Combining eq. (17) with the fact that $d\phi_{\mathbf{x}}(\theta_t) \leq d\phi_{\mathbf{x}}^A(\theta_t)$, we get:

$$d\phi_{\mathbf{x}}(\theta_t) \leq \left( \frac{p}{\beta} - \langle \theta_t - \mathbf{x}, \nabla\mathcal{L}(\theta_t; D) \rangle \right) \cdot dt + \frac{\sqrt{2}}{\beta} \cdot \langle \theta_t - \mathbf{x}, dW_t \rangle. \tag{18}$$

Furthermore, by linearity of expectation we have the following:

$$\mathbb{E}\left[d\phi_{\mathbf{x}}(\theta_t)\right] \leq \frac{p}{\beta} \cdot dt - \mathbb{E}\left[\langle \theta_t - \mathbf{x}, \nabla\mathcal{L}(\theta_t; D) \rangle\right] \cdot dt + \frac{\sqrt{2}}{\beta} \cdot \mathbb{E}\left[\langle \theta_t - \mathbf{x}, dW_t \rangle\right]$$

$$\Leftrightarrow \mathbb{E}\left[\langle \theta_t - \mathbf{x}, \nabla\mathcal{L}(\theta_t; D) \rangle\right] \cdot dt \leq \frac{p}{\beta} \cdot dt + \frac{\sqrt{2}}{\beta} \cdot \mathbb{E}\left[\langle \theta_t - \mathbf{x}, dW_t \rangle\right] - \mathbb{E}\left[d\phi_{\mathbf{x}}(\theta_t)\right] = \frac{p}{\beta} \cdot dt - \mathbb{E}\left[d\phi_{\mathbf{x}}(\theta_t)\right] \tag{19}$$

The last equality in eq. (19) follows from the following observation:

$$\mathbb{E}\left[\langle \theta_t - \mathbf{x}, dW_t \rangle\right] = \mathbb{E}\left[\langle \theta_t, dW_t \rangle\right] - \mathbb{E}\left[\langle \mathbf{x}, dW_t \rangle\right]$$
$$= \mathbb{E}\left[\langle \theta_t, dW_t \rangle\right] \tag{20}$$

$$= -\beta \cdot \mathbb{E}\left[\left\langle \int_{\tau=0}^{t} \nabla\mathcal{L}(\theta_\tau; D) \cdot d\tau, dW_t \right\rangle\right] + \sqrt{2} \cdot \mathbb{E}\left[\int_{\tau=0}^{t} \langle dZ_\tau, dW_t \rangle\right] \tag{21}$$

$$= \sqrt{2} \cdot \mathbb{E}\left[\left\langle \mathbb{E}\left[\int_{\tau=0}^{t} dZ_\tau \,\middle|\, dW_t\right], dW_t \right\rangle\right] = 0, \tag{22}$$

where in eq. (21), $Z_\tau$ is the standard Brownian motion, and in eq. (22) we used the law of iterated expectations and the fact that in the Ito integral, $dW_t$ is independent of $\{W_\tau\}_{0\leq\tau\leq t}$.

By convexity of the loss function $\mathcal{L}$ we have:

$$\mathcal{L}(\theta_t; D) - \mathcal{L}(\mathbf{x}; D) \leq \langle \theta_t - \mathbf{x}, \nabla\mathcal{L}(\theta_t; D) \rangle. \tag{23}$$

Combining eq. (19) and eq. (23) we have the following:

$$\mathbb{E}\left[\left(\mathcal{L}(\theta_t; D) - \mathcal{L}(\mathbf{x}; D)\right) dt\right] \leq p \cdot \frac{dt}{\beta} - \mathbb{E}\left[d\phi_{\mathbf{x}}(\theta_t)\right] \tag{24}$$

Let $a \leq b$. Integrating eq. (24) from $[a, b]$, we have the following:

$$\mathbb{E}\left[\int_{t=a}^{b} (\mathcal{L}(\theta_t; D) - \mathcal{L}(\mathbf{x}; D)) \, dt\right] \leq p \cdot \int_{t=a}^{b} \frac{dt}{\beta} + \mathbb{E}\left[\int_{t=b}^{a} d\phi_{\mathbf{x}}(\theta_t)\right]$$

$$= p \cdot \int_{t=a}^{b} \frac{dt}{\beta} + \mathbb{E}\left[\phi_{\mathbf{x}}(\theta_a) - \phi_{\mathbf{x}}(\theta_b)\right]$$

$$= \frac{p \cdot (b - a)}{\beta} + \mathbb{E}\left[\frac{1}{2\beta} \|\theta_a - \mathbf{x}\|_2^2 - \frac{1}{2\beta} \|\theta_b - \mathbf{x}\|_2^2\right] \quad (25)$$

by the definition of the potential $\phi_{\mathbf{x}}(\theta_t) = \frac{1}{2\beta} \|\theta_t - \mathbf{x}\|_2^2$.

Consider two non-negative real numbers $k$ and $\gamma$ such that $\gamma < k$. Define

$$S_{k,\gamma} = \frac{1}{k + \gamma} \int_{t=T-k}^{T} \mathcal{L}(\theta_t; D) dt \quad \text{and} \quad \bar{S}_k = \lim_{\gamma \to 0} S_{k,\gamma}.$$

That is, $\bar{S}_k$ is the average value of $\mathcal{L}(\theta_t; D)$ over the interval $[T - k, T]$, and in particular $\bar{S}_0 = \mathcal{L}(\theta_T; D)$. We have the following:

$$\mathbb{E}\left[k \cdot S_{k-\gamma,\gamma}\right] = \mathbb{E}\left[\int_{t=T-(k-\gamma)}^{T} \mathcal{L}(\theta_t; D) dt\right] = \mathbb{E}\left[\int_{t=T-k}^{T} \mathcal{L}(\theta_t; D) dt\right] - \mathbb{E}\left[\int_{t=T-k}^{T-(k-\gamma)} \mathcal{L}(\theta_t; D) dt\right]$$

$$= \mathbb{E}\left[(k + \gamma) \cdot S_{k,\gamma}\right] - \mathbb{E}\left[\int_{t=T-k}^{T-(k-\gamma)} \mathcal{L}(\theta_t; D)\right] dt \quad (26)$$

To upper bound $-\mathbb{E}\left[\int_{t=T-k}^{T-(k-\gamma)} \mathcal{L}(\theta_t; D) dt\right]$ above, we use eq. (25) as follows, while setting $a = T - k$ and $b = T$.

$$-\mathbb{E}\left[\mathcal{L}(\mathbf{x}; D)\right] \leq \frac{p}{\beta} + \frac{1}{2\beta(b - a)} \mathbb{E}\left[\|\theta_a - \mathbf{x}\|_2^2\right] - \frac{1}{b - a} \mathbb{E}\left[\int_{a}^{b} \mathcal{L}(\theta_t; D) dt\right]$$

$$= \frac{p}{\beta} + \frac{1}{2\beta \cdot k} \mathbb{E}\left[\|\theta_{T-k} - \mathbf{x}\|_2^2\right] - \frac{k + \gamma}{k} \mathbb{E}\left[S_{k,\gamma}\right]$$

This implies that

$$-\mathbb{E}\left[\int_{t=T-k}^{T-(k-\gamma)} \mathcal{L}(\theta_t; D) dt\right] \leq \frac{p\gamma}{\beta} + \frac{1}{2\beta \cdot k} \mathbb{E}\left[\int_{t=T-k}^{T-(k-\gamma)} \|\theta_{T-k} - \theta_t\|_2^2 dt\right] - \left(1 + \frac{\gamma}{k}\right) \cdot \gamma \cdot \mathbb{E}\left[S_{k,\gamma}\right]$$

$$\leq \frac{p\gamma}{\beta} + \frac{\gamma \cdot \|\mathcal{C}\|_2^2}{2\beta \cdot k} - \left(1 + \frac{\gamma}{k}\right) \cdot \gamma \cdot \mathbb{E}\left[S_{k,\gamma}\right] \quad (27)$$

Plugging in eq. (27) into eq. (26) we have the following:

$$\mathbb{E}\left[k \cdot S_{k-\gamma,\gamma}\right] \leq \mathbb{E}\left[\left(k - \frac{\gamma^2}{k}\right) S_{k,\gamma}\right] + \frac{p\gamma}{\beta} + \frac{\gamma \cdot \|\mathcal{C}\|_2^2}{2\beta \cdot k}$$

Dividing by $k$, we get

$$\mathbb{E}\left[S_{k-\gamma,\gamma}\right] \leq \mathbb{E}\left[\left(1 - \frac{\gamma^2}{k^2}\right) S_{k,\gamma}\right] + \frac{p\gamma}{\beta k} + \frac{\gamma \cdot \|\mathcal{C}\|_2^2}{2\beta \cdot k^2}$$

Rearranging the terms, we get

$$\mathbb{E}\left[S_{k-\gamma,\gamma} - \left(1 - \frac{\gamma^2}{k^2}\right) S_{k,\gamma}\right] \leq \frac{p\gamma}{\beta k} + \frac{\gamma \cdot \|\mathcal{C}\|_2^2}{2\beta \cdot k^2}$$

This gives the following set of implications

$$\mathbb{E}\left[\lim_{\gamma \to 0} \frac{S_{k-\gamma,\gamma} - S_{k,\gamma}}{\gamma}\right] \underset{(*)}{\leq} \lim_{\gamma \to 0} \mathbb{E}\left[\frac{S_{k-\gamma,\gamma} - S_{k,\gamma}}{\gamma}\right] \leq \frac{p}{\beta k} + \frac{\|\mathcal{C}\|_2^2}{2\beta \cdot k^2}$$

$$\Rightarrow \frac{d}{dc}\mathbb{E}\left[\bar{S}_{k-c}\right] \leq \frac{p}{\beta k} + \frac{\|\mathcal{C}\|_2^2}{2\beta \cdot k^2}$$

$$\Rightarrow \mathbb{E}\left[\bar{S}_0\right] \leq \mathbb{E}\left[\bar{S}_k\right] + \frac{p}{\beta} + \frac{\|\mathcal{C}\|_2^2}{2\beta \cdot k}. \tag{28}$$

Where the inequality $(*)$:

$$\mathbb{E}\left[\lim_{\gamma \to 0} \frac{S_{k-\gamma,\gamma} - S_{k,\gamma}}{\gamma}\right] \leq \lim_{\gamma \to 0} \mathbb{E}\left[\frac{S_{k-\gamma,\gamma} - S_{k,\gamma}}{\gamma}\right] \tag{29}$$

follows because of the following argument. Using the definition of $\frac{S_{k-\gamma,\gamma} - S_{k,\gamma}}{\gamma}$ we get:

$$\frac{S_{k-\gamma,\gamma} - S_{k,\gamma}}{\gamma} = \frac{1}{k\gamma} \int_{t=T-k+\gamma}^{T} \mathcal{L}(\theta_t; D)dt - \frac{1}{(k+\gamma)\gamma} \int_{t=T-k}^{T} \mathcal{L}(\theta_t; D)dt$$

$$= \frac{1}{k(k+\gamma)} \int_{t=T-k+\gamma}^{T} \mathcal{L}(\theta_t; D)dt - \frac{1}{\gamma(k+\gamma)} \int_{t=T-k}^{T-k+\gamma} \mathcal{L}(\theta_t; D)dt$$

$$\geq \frac{k-\gamma}{k(k+\gamma)} \min_t \mathcal{L}(\theta_t; D) - \frac{1}{(k+\gamma)} \max_t \mathcal{L}(\theta_t; D). \tag{30}$$

Since $\theta_t \in \mathcal{C}$ for all $t$, and $\mathcal{L}$ is $L$-Lipschitz, we get that $\mathcal{L}(\theta_t, D)$ is lower and upper bounded by some constants for all $t$. Using eq. (30), this implies that $\frac{S_{k-\gamma,\gamma} - S_{k,\gamma}}{\gamma}$ is lower bounded by some constant for all $\gamma < k$. In other words, we can apply Fatou's lemma to upper bound the expectation of the limit by the limit of the expectation, hence the inequality $(*)$.

Now setting $k = T$ in eq. (28), we get:

$$\mathbb{E}\left[\bar{S}_0\right] \leq \mathbb{E}\left[\bar{S}_T\right] + \frac{p}{\beta} + \frac{\|\mathcal{C}\|_2^2}{2\beta T}. \tag{31}$$

If we set $a = 0, b = T$ in eq. (25) and then divide by $T$ we have:

$$\mathbb{E}\left[\bar{S}_T\right] - \mathcal{L}(\mathbf{x}; D) \leq \frac{p}{\beta} + \mathbb{E}\left[\frac{1}{2\beta T}\|\theta_0 - \mathbf{x}\|_2^2 - \frac{1}{2\beta T}\|\theta_T - \mathbf{x}\|_2^2\right] \leq \frac{p}{\beta} + \frac{\|\mathcal{C}\|_2^2}{2\beta T}. \tag{32}$$

Combining eq. (31) and eq. (32) and setting $\mathbf{x} = \theta^*$ where $\theta^*$ is any minimizer over $\mathcal{C}$, we get:

$$\mathbb{E}\left[\mathcal{L}(\theta_T; D) - \mathcal{L}(\theta^*; D)\right] \leq \frac{2p}{\beta} + \frac{\|\mathcal{C}\|_2^2}{\beta T}. \tag{33}$$

We now need to choose $\beta, T$ to minimize the above bound while satisfying privacy. By setting $\Delta = 2L/n$ in Lemma C.1 and setting $\alpha = 1 + \frac{2\log(1/\delta)}{\varepsilon}$ in Fact B.5, we get that as long as $\frac{2\log(1/\delta)}{\varepsilon} \geq 1$ to satisfy $(\varepsilon, \delta)$-differential privacy it suffices if:

$$\frac{4\log(1/\delta)L^2\beta^2 T}{\varepsilon n^2} \leq \frac{\varepsilon}{2} \implies \beta \leq \frac{\varepsilon n}{L\sqrt{8T\log(1/\delta)}}. \tag{34}$$

Plugging $\beta = \frac{\varepsilon n}{L\sqrt{8T\log(1/\delta)}}$ into eq. (33) we get:

$$\mathbb{E}\left[\mathcal{L}(\theta_T; D) - \mathcal{L}(\theta^*; D)\right] \leq \frac{4pL\sqrt{2T\log(1/\delta)}}{\varepsilon n} + \frac{2L\sqrt{2\log(1/\delta)}\|\mathcal{C}\|_2^2}{\sqrt{T}\varepsilon n}. \tag{35}$$

The above bound is optimized by choosing $T = \frac{\|\mathcal{C}\|_2^2}{2p}$, giving the theorem statement. $\square$

*Proof of Theorem F.2.* Note that $\beta_t = \frac{d}{dt}B_t$ by definition, and let $\phi_t = e^{mB_t}\|\theta_t - \theta^*\|_2^2$. Similarly to eq. (19), by Ito's Lemma we have:

$$d\phi_t \leq m\beta_t e^{mB_t}\|\theta_t - \theta^*\|_2^2 dt - e^{mB_t}\left\langle \theta^* - \theta_t, -B_t\nabla\mathcal{L}(\theta_t; D)dt + \frac{\sqrt{2}}{B_t}dW_t\right\rangle + pe^{mB_t}dt$$

Taking expectation on both sides,

$$\mathbb{E}\left[d\phi_t\right] \leq \beta_t e^{mB_t}\mathbb{E}\left[m\|\theta_t - \theta^*\|_2^2 + \langle\theta^* - \theta_t, \nabla\mathcal{L}(\theta_t; D)\rangle\right]dt + pe^{mB_t}dt. \tag{36}$$

By $2m$-strong convexity we have

$$\mathcal{L}(\theta^*; D) - \mathcal{L}(\theta_t; D) \geq \langle\nabla\mathcal{L}(\theta_t; D), \theta^* - \theta_t\rangle + m\|\theta_t - \theta^*\|_2^2. \tag{37}$$

Note that $de^{mB_t} = m\beta_t e^{mB_t}dt$. Then plugging eq. (37) into eq. (36) we get:

$$\mathbb{E}\left[d\phi_t\right] \leq \beta_t e^{mB_t}\left[\mathcal{L}(\theta^*; D) - \mathbb{E}\left[\mathcal{L}(\theta_t; D)\right]\right]dt + pe^{mB_t}dt \tag{38}$$

This implies that

$$\frac{1}{m} \left[ \mathbb{E} \left[ \mathcal{L}(\theta_t; D) \right] - \mathcal{L}(\theta^*; D) \right] de^{mB_t} \leq -\mathbb{E} \left[ d\phi_t \right] + pe^{mB_t} dt. \tag{39}$$

Now, integrating eq. (38) from $t = 0$ to $T$ and noting that $\phi_t \geq 0$ for all $t$ we get:

$$\int_0^T [\mathbb{E} \left[ \mathcal{L}(\theta_t; D) \right] - \mathcal{L}(\theta^*; D)] de^{mB_t} \leq m\phi_0 + mp \int_0^T e^{mB_t} dt. \tag{40}$$

By Jensen's inequality,

$$(e^{mB_T} - 1)\mathcal{L}(\theta^{\mathsf{priv}}; D) \leq \int_0^T \mathcal{L}(\theta_t; D) de^{mB_t}. \tag{41}$$

Plugging eq. (41) into eq. (40) and using linearity of expectation we have:

$$\mathbb{E} \left[ \mathcal{L}(\theta^{\mathsf{priv}}; D) \right] - \mathcal{L}(\theta^*; D) \leq \frac{m\phi_0 + mp \int_0^T e^{mB_t} dt}{e^{mB_T} - 1}. \tag{42}$$

We now determine what choice of $a$ as a function of $T$ ensures privacy when $\beta_t = t^a$. Our final choice of $a$ will satisfy $a \geq 0$ for all sufficiently large $n$. Recall from Lemma C.1 that for $(\varepsilon, \delta)$-differential privacy, since we have $L$-Lipschitzness it suffices if:

$$\frac{4L^2 \log(1/\delta) \int_0^T \beta_t^2 dt}{\varepsilon n^2} \leq \varepsilon$$

$$\Leftrightarrow \int_0^T t^{2a} dt \leq \frac{\varepsilon^2 n^2}{4L^2 \log(1/\delta)}$$

$$\Leftrightarrow T^{2a+1} \leq \frac{(2a+1)\varepsilon^2 n^2}{4L^2 \log(1/\delta)} \tag{43}$$

Note that we use the assumption $a \geq 0$ to show the integral is equal to $T^{2a+1}$ rather than diverging. For eq. (43) to hold, assuming $a \geq 0$, we have $2a + 1 \geq 1$, so letting $a = \log_{\max\{2,T\}} \left( \frac{\varepsilon n}{2L\sqrt{\log(1/\delta)}} \right) - \frac{1}{2}$ suffices. We will eventually choose $T$ that is at most 2 for sufficiently large $n$, which implies that $a \geq 0$ for all sufficiently large $n$ as promised. We now wish to evaluate from eq. (42) the integral:

$$\int\limits_0^T \exp(mB_t)dt = \int\limits_0^T \exp(\frac{mt^{a+1}}{a+1})dt = \int\limits_0^T \exp\left(\frac{mt^{\log_{\max\{2,T\}}\left(\frac{\varepsilon n}{2L\sqrt{\log(1/\delta)}}\right)}\sqrt{t}}{a+1}\right)dt \tag{44}$$

$$\leq \int\limits_0^T \exp\left(\frac{m\varepsilon n\sqrt{t}}{2(a+1)L\sqrt{\log(1/\delta)}}\right)dt. \tag{45}$$

Let $c = \frac{m\varepsilon n}{2(a+1)L\sqrt{\log(1/\delta)}}$, and $x = c\sqrt{t}$. Then $\frac{dx}{dt} = \frac{c}{2\sqrt{t}} = \frac{c^2}{2x}$. So the last integral in eq. (44) can be upper bounded as:

$$\int\limits_0^T \exp(c\sqrt{t})dt = \frac{2}{c^2}\int\limits_0^{c\sqrt{T}} xe^x dx = 2\cdot\frac{e^{c\sqrt{T}}(c\sqrt{T}-1)+1}{c^2} \leq 2\cdot\frac{\sqrt{T}(e^{c\sqrt{T}}-1)}{c} \tag{46}$$

Where the last inequality follows from the inequality $e^x \geq 1+x$. We also have that the denominator of eq. (42) is equal to $e^{mB_t}-1 = e^{c\sqrt{T}}-1$. So plugging eq. (44), eq. (46) into eq. (42) we have:

$$\mathbb{E}\left[\mathcal{L}(\theta^{\mathsf{priv}};D)\right] - \mathcal{L}(\theta^*;D) \leq \frac{m\phi_0}{e^{c\sqrt{T}}-1} + \frac{2mp\sqrt{T}}{c} \tag{47}$$

If we choose $T = \frac{\log^2(R+1)}{c^2}$ and plug in the expression for $c$, eq. (47) becomes:

$$\mathbb{E}\left[\mathcal{L}(\theta^{\mathsf{priv}};D)\right] - \mathcal{L}(\theta^*;D) \leq \frac{m\phi_0}{R} + \frac{2\log(R+1)}{c^2} = \frac{m\phi_0}{R} + \frac{8p(a+1)L^2\log(1/\delta)\log(R+1)}{m\varepsilon^2 n^2}. \tag{48}$$

Note that $\phi_0 = \|\theta_0 - \theta^*\|_2^2 \leq \|\mathcal{C}\|_2^2 \leq \frac{L^2}{m^2}$. So to complete the proof, we can choose

$$R = \frac{m^2\varepsilon^2 n^2 p\phi_0}{2L^2\log(1/\delta)} \leq \frac{p\varepsilon^2 n^2}{2\log(1/\delta)}, \tag{49}$$

Which causes the second term in eq. (48) to be larger than the first term and gives the theorem statement. $\qquad\square$

*Proof of Theorem F.4.* We again choose $\beta = \frac{\varepsilon n}{L\sqrt{8T\log(1/\delta)}}$ as in Theorem F.1. Then combining Lemma F.3 with eq. (35), we get the following bound

$$\mathbb{E}_{\theta^{\mathsf{priv}}}[\mathsf{Risk}_{\mathsf{SCO}}(\theta^{\mathsf{priv}})] \leq \sqrt{T}\left(\frac{4pL\sqrt{2\log(1/\delta)}}{\varepsilon n} + \frac{\sqrt{2}\varepsilon L}{\sqrt{\log(1/\delta)}}\right) + \frac{1}{\sqrt{T}}\left(\frac{2L\sqrt{2\log(1/\delta)}\|\mathcal{C}\|_2^2}{\varepsilon n}\right) \tag{50}$$

To minimize the right hand side of the above inequality, we choose $T = \Theta(\min\{\frac{\|\mathcal{C}\|_2^2}{p}, \frac{\log(1/\delta)\|\mathcal{C}\|_2^2}{\varepsilon^2 n}\})$. The result follows. $\qquad\square$

