# OpenReview forum: "On the Universality of Langevin Diffusion for Private Euclidean (Convex) Optimization"
_ICLR.cc/2023/Conference — Submitted to ICLR 2023_

### Official Review · Reviewer_ymJu · 2022-10-22

**Confidence:** 3
**Correctness:** 3
**Technical Novelty And Significance:** 3
**Empirical Novelty And Significance:** 3
**Recommendation:** 5

**Clarity, Quality, Novelty And Reproducibility:**

The paper is not written in a very clear way as I explained in the weaknesses previously. It is not clear to me when certain assumptions are made and when they are not. The quality of the presentation should be improved.

**Strength And Weaknesses:**

The paper is a solid theory paper, and the authors have provided detailed and rigorous treatment in the relatively long appendix. The authors also seem to be quite familiar with the relevant literature. The topic is important and worth investigating.

However, I have some concerns about the assumptions and especially how the paper is presented.

(1) The assumptions are often confusing. For example, on page 2, when the authors introduce the problem description, one considers a constrained set $\mathcal{C}\subseteq\mathbb{R}^{d}$. However, it is not clear to me when $\mathcal{C}=\mathbb{R}^{d}$ and when it is a proper subset of $\mathbb{R}^{d}$. Moreover, if it is a proper subset of $\mathbb{R}^{d}$, what is the assumption on $\mathcal{C}$? For another example, the authors mentioned on page 3, that "Depending on the problem context, we make additional assumptions like $m$-strong convexity..." I am perfectly fine with this. However, it is not clear when you make such an assumption and when you do not. For example, in the statement of Lemma 1.1., you didn't mention the strong convexity assumption, so I suppose it is not used. But then, in Theorem 1.2., $m$ appears in the equation, does that mean you used this $m$-strongly convex assumption? If so, you should mention it. If not, you should mention what $m$ means in Theorem 1.2.

(2) It seems that the paper is hard to read unless you go back and forth between the main body and the appendix. For example, in Theorem 1.2., it uses Algorithm 3, but you cannot find Algorithm 3 in the main paper at all.

(3) Some of the assumptions might be too strong. For example, the authors assumed that the loss function is $L$-Lipschitz. That assumption is quite strong when $\mathcal{C}=\mathbb{R}^{d}$, because for the Langevin literature that I am familiar with, people usually just assume (strong) convexity plus smoothness, but not the Lipschitz assumption. If your $\mathcal{C}$ is compact, you need to make an assumption and make it clear in the main body of the paper. For instance, when I look at the statement of Theorem 1.3., it seems to me that $\mathcal{C}$ needs to be bounded. But when I'm looking at Lemma 1.1., I don't know whether you need assumption on $\mathcal{C}$ or not.

(4) I am a bit confused with equation (2), about "Projected" Langevin diffusion on page 3. I understand that for Langevin algorithm on a constrained domain, you can project it back to the constrained set if the algorithm exits the domain. However, in equation (2), you are considering a continuous-time dynamics. In Bubeck et al., they showed that (discrete-time) projected Langevin algorithm, in the continuous-time limit, corresponds to a Langevin diffusion that is reflected when it touches on the boundary. You are considering a continuous-time SDE in equation (2), that makes me wonder why it is not a reflected SDE, but rather involves a projection map?

(5) The fact that the authors are considering a "continuous-time" algorithm, instead of a discrete time algorithm, makes the contributions of the paper a bit weak.

(6) Langevin algorithm is used in the optimization literature usually when the objective is non-convex, so that the injected Gaussian noise can help the algorithm escape local minima; see e.g. Raginsky et al. (2017). It is not clear to me when for convex optimization, one needs to use a Langevin algorithm.

**Summary Of The Paper:**

This paper studies differentially private empirical risk minimization (DP-ERM) and differentially private stochastic (convex) optimization (DP-SCO). The authors consider the Langevin diffusion, which is a popular continuous-time algorithm, and show that Langevin diffusion has optimal privacy and utility trade-offs for both DP-ERM and DP-SCO under $\varepsilon$-DP and $(\varespilon,\delta)$-DP both for convex and strongly convex loss functions. They also obtain new uniform stability results for Langevin diffusion that are time and dimension independent.

**Summary Of The Review:**

The paper studies an important and interesting topic. It is a solid theory paper. However, it has quite a few weaknesses as I mentioned previously.

---

> ### Author Response · Authors · 2022-11-09
> **Response to Reviewer ymJu**
>
> Thank you for the review! We are currently preparing a revision which will hopefully address some of the concerns raised. Meanwhile, please see below responses to some of the points raised in the review.
>
> > The assumptions are often confusing. For example, on page 2, when the authors introduce the problem description, one considers a constrained set $\mathcal{C} \subseteq \mathbb{R}^p$. However, it is not clear to me when $\mathcal{C} = \mathbb{R}^p$ and when it is a proper subset of $\mathbb{R}^p$. Moreover, if it is a proper subset of , what is the assumption on ?
>
> We apologize for the confusion! In all settings in Table 1 we consider when $\mathcal{C}$ is a convex subset of $\mathbb{R}^p$ with bounded diameter (which is the standard assumption in the DP-ERM/SCO literature). In the revision we will remove the possibility for equality with $\mathbb{R}^p$ when defining $\mathcal{C}$ in the introduction and state that this assumption holds for all our upper bounds.
>
> > For another example, the authors mentioned on page 3, that "Depending on the problem context, we make additional assumptions like $m$-strong convexity..." I am perfectly fine with this. However, it is not clear when you make such an assumption and when you do not.
>
> Apologies for the confusion; throughout the paper we only use $m$ to mean a strong convexity parameter, though to be clearer we will revise the paper to state explicitly that we assume $m$-strong convexity in each formal statement where m appears.
>
> > For example, the authors assumed that the loss function is $L$-Lipschitz. That assumption is quite strong when $\mathcal{C} = \mathbb{R}^p$, because for the Langevin literature that I am familiar with, people usually just assume (strong) convexity plus smoothness, but not the Lipschitz assumption
>
> Please see our first response. Restating it here, in all settings in Table 1 we assume $\mathcal{C}$ is a proper subset of $\mathbb{R}^p$ with bounded diameter, and then strong convexity within $\mathcal{C}$ and Lipschitzness within $\mathcal{C}$ are not incompatible.
>
> > I am a bit confused with equation (2), about "Projected" Langevin diffusion on page 3. I understand that for Langevin algorithm on a constrained domain, you can project it back to the constrained set if the algorithm exits the domain. However, in equation (2), you are considering a continuous-time dynamics. In Bubeck et al., they showed that (discrete-time) projected Langevin algorithm, in the continuous-time limit, corresponds to a Langevin diffusion that is reflected when it touches on the boundary. You are considering a continuous-time SDE in equation (2), that makes me wonder why it is not a reflected SDE, but rather involves a projection map?
>
> We presented (2) as a “projected” diffusion rather than a reflected one to hopefully ease a reader who is familiar with private optimization but not with stochastic calculus into the definition of (2). To our understanding, the “projected” diffusion we define is identical to the reflected SDE, although admittedly this statement requires a more formal proof. We will change the presentation to use the reflected Brownian motion instead, and encourage readers unfamiliar with stochastic calculus to think of it as the limit of projected SGD rather than try to parse the details of the SDE. The proofs in Appendix F go through under either presentation of (2), and the rest of the paper is unaffected.
>
> > The fact that the authors are considering a "continuous-time" algorithm, instead of a discrete time algorithm, makes the contributions of the paper a bit weak.
>
> We agree that it is important to look at the discrete algorithms, and in practice of course one should use DP-SGD which is very well understood. The study of continuous-time algorithms in this paper was primarily for the sake of giving a unified viewpoint of the different DP optimization settings.
>
> > Langevin algorithm is used in the optimization literature usually when the objective is non-convex, so that the injected Gaussian noise can help the algorithm escape local minima; see e.g. Raginsky et al. (2017). It is not clear to me when for convex optimization, one needs to use a Langevin algorithm.
>
> In the non-private setting, we agree there is no reason to use Langevin diffusion instead of an unnoised gradient descent/drift for convex losses. In differentially private optimization, however, it is necessary to add noise to the gradients (as in DP-SGD or Langevin diffusion) for the sake of preserving privacy.

---

### Official Review · Reviewer_J76Y · 2022-10-24

**Confidence:** 3
**Correctness:** 3
**Technical Novelty And Significance:** 3
**Empirical Novelty And Significance:** 1
**Recommendation:** 5

**Clarity, Quality, Novelty And Reproducibility:**

Paper is not clear at all and to be able to understand one should go back and forth and it feels like the contributions are lost. It somehow feels like it is written to confuse the reader. It is too long for a conference paper. This paper has some novel aspects even though some ideas are directly borrowed. The mathematical results can be reproduced.

**Strength And Weaknesses:**

Strength:
- This is a very interesting problem to look at even though this is not the first time that DP is viewed through the Langevin dynamic.
- Provides a comprehensive background on the topic.
------------------------------------------------------------------------
Weakness:
- This paper can be hard to follow. The main body of the paper is not self-sufficient and one should constantly go to the appendix to understand the paper. Perhaps it could have been a bit more organized.
- This paper is way too long for a conference paper.  And it feels like the main contribution is lost in the notation and the amount of unnecessary information provided in the paper.
- It feels like the main idea stems from one of the recent papers on the use of Langevin diffusion for DP (DP guarantees for SGLD by Ryffel), without having any sort of comparison for them. Notations, use of Renyi divergence and etc are borrowed from the same paper.
- Proofs seem to be correct as they follow the properties of LD and straightforward techniques in DP.
- Due to the similarity of techniques (use of LD in this paper and the one by Ryffel), comparison/discussion on both these methods is required. Since some techniques are borrowed, how is this work different aside from being applied to ERM and SCO. Techniques seem to be very similar.
- What can say about the sensitivity under the LD?
- Are strong convexity a necessary condition? If this assumption was not made, how would the bound change?
- I understand this paper is meant to be understood as theoretical but some practical implications of that could be helpful.


**Summary Of The Paper:**

This paper addresses differential privacy using the Langevin dynamic and claims to achieve the optimal utility-privacy trade-off for DP-ERM and DP-SCO. This paper provides bounds for excess risk on the \epsilon-DP-SCO.

**Summary Of The Review:**

This paper provides bounds on the DP-ERM and DP-SCO with the use of the Langevin dynamic. Seems like, with techniques borrowed from the literature, this paper provides bounds that claim to be "tight" specifically for \epsilon-DP.

---

> ### Author Response · Authors · 2022-11-09
> **Response to Reviewer J76Y**
>
> Thank you for the review. We are currently preparing a revision that will hopefully address some of your concerns. Meanwhile, please see responses below to some points made in the review.
>
> >This paper can be hard to follow. The main body of the paper is not self-sufficient and one should constantly go to the appendix to understand the paper. Perhaps it could have been a bit more organized.
>
> We will attempt to add more details where possible to the introduction in the revision.
>
> > It feels like the main idea stems from one of the recent papers on the use of Langevin diffusion for DP (DP guarantees for SGLD by Ryffel), without having any sort of comparison for them. Notations, use of Renyi divergence and etc are borrowed from the same paper.
>
> We do not believe our paper’s main ideas draw anything from Ryffel et al. that did not exist in other works already. The connections between Langevin diffusion (which we note is slightly different from SGLD) and DP date back to at least the work of Vempala and Wibisono (2019) and Ganesh and Talwar (2020). Renyi divergence/RDP and the associated tools and notation are drawn from van Erven and Harremoes (2012) and Mironov (2017). Our analysis of LD as a continuous optimization method is based on techniques in Wilson (2018). Also note that the techniques of Ryffel et al. are heavily reliant on strong convexity, whereas we frequently operate in the convex setting, so we cannot draw from their techniques.
>
> We will include a mention of Ryffel et al. in our comparison to Chourasia et al., since Ryffel et al. directly improves upon that work. However, both those works’ goals, namely an improved privacy analysis for full/mini-batch SGD in the strongly convex setting, as well as their techniques are largely orthogonal to ours, so a more extended comparison does not feel warranted.
>
> > Due to the similarity of techniques (use of LD in this paper and the one by Ryffel), comparison/discussion on both these methods is required. Since some techniques are borrowed, how is this work different aside from being applied to ERM and SCO. Techniques seem to be very similar.
>
> Per the reasons in the previous response, we do not believe an extensive comparison with Ryffel et al. is warranted, as both the goals and techniques of Ryffel et al. are orthogonal to ours.
>
> > What can say about the sensitivity under the LD?
>
> Could you clarify what is meant by sensitivity? If you mean gradient sensitivity to the database, this is given by the Lipschitz assumption. If you mean uniform stability, we show uniform stability bounds for LD in the paper.
>
> > Are strong convexity a necessary condition? If this assumption was not made, how would the bound change?
>
> As per Table 1, we show bounds both under convexity (which is a standard assumption in the DP-ERM/SCO literature) and strong convexity in all settings. For our uniform stability bound which assumes strong convexity, in the convex case we can apply it by adding a mild regularizer.
>
> > I understand this paper is meant to be understood as theoretical but some practical implications of that could be helpful.
>
> For approximate DP, in practice of course one should use a discrete algorithm such as DP-SGD which is known to be optimal and generally well understood. However, our unified framework does give new insights on this algorithm (e.g. how it converges in an optimization sense but not in a sampling sense to the best of our knowledge, we are the first to observe that DP-SGD obtains the same loss bound as its stationary distribution, but does not converge to it in statistical distance. This can help us understand DP-SGD's behavior in practice). For pure DP, the insight is new guarantees and easy-to-derive uniform stability properties for the exponential mechanism, which is a widely used primitive in the DP space that is in some settings efficient to implement.

---

> > ### Comment · Reviewer_J76Y · 2022-11-29
> > **Thank you for your response!**
> >
> > I'd like to thank the authors for their response. Even though I agree with other reviewers regrading the discretization and issues stemming from that, the problem itself is challenging and interesting. Using Langevin in differential privacy introduces a very interesting approach. However, it is not the first time for Langevin dynamic to appear in the context of DP. I do believe there is merits to their techniques but as mentioned by others I am not fully convinced of improvements due to this generalization. I thus keep my score as it was.

---

### Official Review · Reviewer_yW9T · 2022-10-25

**Confidence:** 3
**Correctness:** 4
**Technical Novelty And Significance:** 3
**Empirical Novelty And Significance:** 2
**Recommendation:** 6

**Clarity, Quality, Novelty And Reproducibility:**

The paper is in general clearly written.
There is no experiment.

**Strength And Weaknesses:**

Strength:

Although most techniques used in this paper exist in prior work, they are adapted to the specific problem considered here.
Combining these techniques, authors not only recover best known bounds, but also derive new results.
The collection of optimal bounds under various setting manifest that Langevin diffusion is indeed capable to serve as a unified framework to analysis empirical risk and population risk with privacy constraints.

Weakness:
The main weakness is that algorithms 2,3 in this paper are not actually algorithm, because sampling exactly from an Gibbs measure is generally not possible within finite number of operations.
Any implementation necessarily introduce a sampling bias which decrease with computation budget. For example, DP-SGD in (Bassily et al., 2014) uses a fixed step size.
It is still unclear how such bias interfere with bounds provided in this paper.

Another consequence of continuous model is that the effect of stochastic gradient error is totally ignored. For example, DP-SGD in (Bassily et al., 2014) uses a mini-batch for each iteration. However, in continuous time LD, all information of the whole dataset is used at every time point.

Given these mismatch between continuous time LD and real world algorithms, it is not clear how significant results in this paper is to the development of practical differentially private algorithms.

More specifically, it seems all improvement over existing results happens for ε-DP with $T=\infty$ (please correct me if I am wrong), thus it is hard to see whether the improvement is due to better algorithm design or a bonus of infinite computation power.

typos:

In eq 34, ε in denominator should not be square.

**Summary Of The Paper:**

This paper studies the differentially privacy guarantee, empirical risk bound and population risk bound of continuous time Langevin dynamics with tunable temperature and tuning time. For convex loss, the ε-DP result is cited from (Bassily et al., 2014). The ε-DP result for strongly convex function is derived for some iterated exponential mechanism similar to (Bassily et al., 2014). To derive the ε,δ-DP guarantee this paper uses Renyi differentially privacy as a sufficient condition. The Renyi divergence of two Langevin diffusions is derived using standard combinations of post processing theorem and composition theorem.
The population risk for stochastic convex optimization is derived with uniform stability bound similar to (Hardt et al., 2016).
Combining these techniques, authors show various optimal bound under privacy guarantee in a unified model ,i.e. Langevin diffusion.

**Summary Of The Review:**

This paper present continuous Langevin dynamics as a unified approach to the problem of differential privacy. In particular, the bounds derived in this paper recover or exceed the best known bounds in the field. At the same time, the limitation of continuous dynamics and infinite computation budget raise questions on the practical significance.

---

> ### Author Response · Authors · 2022-11-09
> **Response to Reviewer yW9T**
>
> Thank you for the review! We are working on a revision that will hopefully address some of your concerns; in the mean time, please see below responses to some points made in the review.
>
> > Another consequence of continuous model is that the effect of stochastic gradient error is totally ignored. For example, DP-SGD in (Bassily et al., 2014) uses a mini-batch for each iteration. However, in continuous time LD, all information of the whole dataset is used at every time point.
>
> The purpose of this work was to find any object unifying the different DP-ERM/DP-SCO results even if information theoretic; we acknowledge that we did not consider computation cost. We note that most past work in the pure-DP settings and some past works in the approx-DP settings also use some variant of the exponential mechanism, which is in general an information theoretic object, as the basis for their algorithm.
>
> > More specifically, it seems all improvement over existing results happens for ε-DP with $T = \infty$ (please correct me if I am wrong), thus it is hard to see whether the improvement is due to better algorithm design or a bonus of infinite computation power.
>
> When we say $T = \infty$, we effectively mean that we are using the exponential mechanism, which Bassily et al. also used for the previous best epsilon-DP ERM bounds. So that improvement is not an artifact of the computational power used.
>
> The algorithm of Asi et al. giving the best epsilon-DP-SCO bounds requires being able to exactly solve an ERM problem, which is a weaker primitive/easier to approximate than an exponential mechanism.  We would like to remark that there are aspects of our algorithm design that seem to improve on theirs (e.g. their algorithm also operates in rounds but distributes the epsilons differently, the uniform stability bound we prove is a generalization of the one used in their paper). We will mention both these points in the paper for a more fair comparison.
>
> > In eq 34, ε in denominator should not be square.
>
> Thank you for catching this! We will fix this in the revision.

---

> > ### Comment · Reviewer_yW9T · 2022-11-28
> > **Thanks for the response**
> >
> > Thanks for the response and the new revision.
> >
> > It seems author and I and reviewer ymJu all agree that this paper didn't study discretized algorithm, and result for discretized algorithm would be more importance in practice.
> > In the revision, authors claim the improvements come not only from stronger primitives, but also due to generalization of the uniform stability bound. However, I'm not sure as there is currently no evidence that such generalizations can solely lead to any improvements.
> >
> > On the other hand, this paper indeed gives an interesting and unified framework for differential privacy.
> >
> > Considering the advantages and disadvantages, I decided to maintain my score.

---

### Official Review · Reviewer_wZ3h · 2022-10-25

**Confidence:** 4
**Correctness:** 4
**Technical Novelty And Significance:** 3
**Empirical Novelty And Significance:** 2
**Recommendation:** 6

**Clarity, Quality, Novelty And Reproducibility:**

This paper provides a framework for DP-ERM/DP-SCO with solid theoretical analysis, which may be applicable to other problems.

**Strength And Weaknesses:**

Strengths:
1. The main contribution of the paper is providing a framework based on Langevin diffusion to analyze privacy/utility trade-offs for DP-ERM and DP-SCO.
2. Algorithmic stability is employed to provide the tight DP-SCO bounds.
3. The paper is well-written and solid in theoretical analysis.

Weaknesses:
1. The comparison of tight bounds for DP-ERM and DP-SCO with related work is lacking(a table may can demonstrate the comparison better).
2. The theorems are fairly restrictive. (i.e. For Theorem 1.2, if $p \ll n$, the improvement from $p^2 \log(n)$ to $p\log(n) + p^2$ is limited; if $p \gg n$, the bound seems to be less meaningful. The assumptions for stability bound are required strongly convex and smooth, which are also strict. The choice of learning rate should satisfy empirical case and population case simultaneously.)
3. The experiment part is missing.

**Summary Of The Paper:**

The paper analyzes achieve trade-offs of privacy and utility for DP-ERM and DP-SCO via Langevin diffusion. Tight DP-SCO bounds for both convex and strongly convex are provided through the lens of algorithmic stability. For non-convex case, a lower bound is provided to show the impossibility of improvement from $\epsilon$-DP to $(\epsilon, \delta)$-DP.

**Summary Of The Review:**

Overall, I think the contribution of this paper is to employ Langevin diffusion to provide a framework to analyze the privacy/utility trade-offs, but it does not show a significant improvement compared with previous work. Additionally, the assumptions for uniform stability seem strict. However, it indeed provides a new thinking of analysis.

---

> ### Author Response · Authors · 2022-11-09
> **Response to Reviewer wZ3h**
>
> Thank you for your review! Please see below responses to some of the points raised in the review.
>
> > The theorems are fairly restrictive. (i.e. For Theorem 1.2, if $p \ll n$, the improvement from $p^2 \log n$ to $p^2 + \log n$ is limited;
>
> While it is true that most existing bounds are already tight up to logarithmic factors, our results make two crucial advancements: i) It is the first work to provide a single algorithm that achieves these bounds for both pure and approx. DP in both ERM and SCO settings, and ii) The SCO bounds we obtain in the pure DP case match the non private SCO bounds when $\epsilon\to \infty$. Prior results do not have this property, and it is a fundamental limitation of those techniques. New tight results like these can be viewed as evidence that our framework is a “correct” one for understanding DP-ERM/SCO problems.
>
> Specific to the comment about Theorem 1.2, note that the bound is tight if $p > \log n$.
>
> > The assumptions for stability bound are required strongly convex and smooth, which are also strict.
>
> We note that no smoothness requirement is needed for any of our final results. For the argument for uniform stability of the exponential mechanism, which starts by analyzing the uniform stability of gradient descent under smoothness, the smoothness required is $1/\eta$-smoothness, which is trivially satisfied when we take the limit as $\eta \to 0$ in our argument. So, our final uniform stability bound for the exponential mechanism doesn’t require smoothness in the end. We apologize if this was not clear from the presentation.
>
> Furthermore, one can enforce strong convexity (and thus uniform stability of the exponential mechanism) on a convex loss by adding a small regularizer, and indeed our epsilon-DP-SCO bound for the convex case does this.
>
> > The choice of learning rate should satisfy empirical case and population case simultaneously.)
>
> We are unsure what is meant by learning rate here, since using Langevin diffusion instead of gradient descent removes the need to choose a learning rate/discretization.

---

### Author Response · Authors · 2022-11-11
**New revision uploaded**

Dear reviewers, thank you again for the thoughtful feedback on the paper. We have uploaded a revision with the changes we promised in our responses to address some of the suggestions and criticisms in the reviews. To make it easier to find these changes, we have highlighted the changes in this revision using red text.

---

### Decision · Program_Chairs · 2023-01-20

**Decision:**

Reject

**Justification For Why Not Higher Score:**

While the paper has some interesting conceptual contribution, it has several shortcomings:
(1) The technical analysis is largely present in prior work (e.g., Bassily et al, Hardt et al, standard analyses of Renyi divergence of diffusion processes). This would on its own not an issue, if it conveyed a conceptually new point --- which is here unclear.
(2) While Langevin diffusion is an appealingly clean algorithm to have the optimal trade-off, it is crucially not a runnable algorithm. It needs to be discretized to be turned into an algorithm one can run, and the authors do not analyze discretization effects (nor consequences to computational complexity).


**Justification For Why Not Lower Score:**

N/A

**Metareview: Summary, Strengths And Weaknesses:**

The paper concerns a new unified viewpoint for DP-ERM and DP-SCO via the Langevin diffusion algorithm. Namely, the authors prove that this (natural) algorithm in a certain precise sense provides the optimal utility/privacy trade-off. The results are proven by combining techniques from Bassily et al, as well as stability-based generalization bounds (e.g., Hardt et al). After some discussion with the authors, the conclusion was that while the paper has some interesting conceptual contribution, it has several shortcomings:
(1) The technical analysis is largely present in prior work (e.g., Bassily et al, Hardt et al, standard analyses of Renyi divergence of diffusion processes). This would on its own not an issue, if it conveyed a conceptually new point --- which is here unclear.
(2) While Langevin diffusion is an appealingly clean algorithm to have the optimal trade-off, it is crucially not a runnable algorithm. It needs to be discretized to be turned into an algorithm one can run, and the authors do not analyze discretization effects (nor consequences to computational complexity).